# Particulate trimethylamine in the summertime Canadian high Arctic lower troposphere

Franziska Köllner[1,2], Johannes Schneider[1], Megan D. Willis[3], Thomas Klimach[1], Frank Helleis[1], Heiko Bozem[2], Daniel Kunkel[2], Peter Hoor[2], Julia Burkart[3], W. Richard Leaitch[4], Amir A. Aliabadi[4,a], Jonathan P.D. Abbatt[3], Andreas B. Herber[5], and Stephan Borrmann[1,2]

[1]Max Planck Institute for Chemistry, Mainz, Germany
[2]Institute for Atmospheric Physics, Johannes Gutenberg University Mainz, Germany
[3]Department of Chemistry, University of Toronto, Canada
[4]Environment Canada, Toronto, Canada
[5]Alfred Wegener Institute for Polar and Marine Research, Bremerhaven, Germany
[a]now at: Environmental Engineering Program, University of Guelph, Guelph, Canada
*Correspondence to:* Franziska Köllner (f.koellner@mpic.de)

**Abstract.** Size-resolved and vertical profile measurements of single particle chemical composition (sampling altitude range 50 - 3000 m) were conducted in July 2014 in the Canadian high Arctic during the aircraft-based measurement campaign NETCARE 2014. We deployed the single particle laser ablation aerosol mass spectrometer ALABAMA (vacuum aerodynamic diameter range approximately 200 - 1000 nm) to identify different particle types and their mixing states. On basis of the single

particle analysis, we found that a significant fraction (23 %) of all analyzed particles (in total: 7412) contained trimethylamine (TMA). Two main pieces of evidence suggest that these TMA-containing particles originated from emissions within the Arctic boundary layer. First, the maximum fraction of particulate TMA occurred in the Arctic boundary layer. Second, compared to particles observed aloft, TMA particles were smaller and less oxidized. Further, air mass history analysis, associated wind data and comparison with measurements of methanesulfonic acid give evidence of a marine-biogenic influence on particulate TMA.

Moreover, the external mixture of TMA-containing particles and sodium and chloride ("Na/Cl-") containing particles, together with low wind speeds suggests particulate TMA results from secondary conversion of precursor gases released by the ocean. In contrast to TMA-containing particles originating from inner-Arctic sources, particles with biomass burning markers (such as levoglucosan and potassium) showed a higher fraction at higher altitudes, indicating long-range transport as their source. Our measurements highlight the importance of natural, marine inner-Arctic sources for composition and growth of summertime

Arctic aerosol.

## 1 Introduction

A remarkable increase in Arctic near-surface air temperature (e.g., Chapman and Walsh, 1993; Serreze et al., 2009) has led to rather drastic changes of several climate parameters, in particular a decreasing sea ice extent of 3.5 % to 4.1 % per decade since 1979 (IPCC, 2014, with further evidence up to 2017 from the National Snow and Ice Data Center, Boulder, Colorado,

https://nsidc.org). Among the processes driving Arctic warming, direct and indirect radiative effects of aerosol particles play a

key role. The impact of aerosol particles on the radiation budget strongly depends on number concentration, size and chemical composition (e.g., Haywood and Boucher, 2000). Different measurements at Arctic sites show a strong annual cycle in these aerosol characteristics (e.g., Tunved et al., 2013; Leaitch et al., 2013; Engvall et al., 2008; Quinn et al., 2007; Rahn and McCaffrey, 1980). Three main processes drive the annual cycle in Arctic aerosol. First, pollution sources within the polar

dome are reduced during summer, since the polar dome surface extent is smaller during summer compared to winter (e.g., Law and Stohl, 2007; Stohl, 2006; Klonecki et al., 2003; Barrie, 1986). Second, efficient wet removal processes in liquid clouds lead to a smaller condensation sink in the summertime Arctic in contrast to wintertime conditions (e.g., Croft et al., 2016a). Third, the substantial change in duration of daylight in Arctic summer leads to increased photochemical processes and increased biological activity, which further result in a higher nucleation potential (e.g., Burkart et al., 2017; Heintzenberg et al., 2017;

Croft et al., 2016b; Wentworth et al., 2016; Leaitch et al., 2013; Kupiszewski et al., 2013; Kawamura et al., 2010, 1996).

To better understand the physical and chemical processes leading to a higher nucleation potential and the frequent appearance of clouds in the summertime Arctic, it is crucial to study emissions of the terrestrial and oceanic biosphere. So far, a few studies have discussed the importance of methanesulfonic acid (MSA), an oxidation product of dimethylsulfide emitted from ocean biomass, to take part in aerosol chemistry in the Arctic (e.g., Croft et al., 2016b; Leaitch et al., 2013; Sharma et al., 2012;

Willis et al., 2016; Mungall et al., 2016). It is further known that marine biota also release certain gas-phase amines, such as trimethylamine (TMA), into the atmosphere (e.g., Ge et al., 2011a; Van Neste et al., 1987; Gibb et al., 1999; Facchini et al., 2008), which subsequently may contribute to aerosol chemistry. Numerous chamber, modeling and field studies at southern latitudes (e.g., Almeida et al., 2013; Kürten et al., 2017; You et al., 2014; Bergman et al., 2015; Müller et al., 2009) have focused on sources, emission rates and gas-to-particle partitioning processes of atmospheric amines. So far, this research has

shown that amines may take part in aerosol chemistry in several ways. These include acid-base reactions to form aminium salts and dissolution in cloud droplets (owing to their high water-solubility) where subsequent acid-base reactions can occur in the aqueous phase (e.g., Glasoe et al., 2015; Dawson et al., 2012; Erupe et al., 2011; Ge et al., 2011b; Jen et al., 2016, 2014; Youn et al., 2015; Yu et al., 2012; Rehbein et al., 2011; Pankow, 2015). Amines compete with ammonia ($NH_3$) in neutralizing acidic aerosol. The base that is favoured by these reactions depends on several parameters, such as acidity of the aerosol, Henry's

law coefficient and the concentration of both substances in the atmosphere (e.g., Pratt et al., 2009b; Barsanti et al., 2009). Amines further may take part in aerosol chemistry via gas-phase oxidation processes leading to the formation of species such as amides, nitramines and imines. The resulting lower volatility products can go on to form secondary organic aerosol (SOA) (e.g., Murphy et al., 2007; Ge et al., 2011b; Angelino et al., 2001).

Despite these considerable advances in studies of atmospheric amines, very little is known about their abundance in Arctic

regions. Scalabrin et al. (2012) reported marine influence on amino acids in Arctic aerosol. Further particle measurements at Mace Head, Ireland have shown the presence of organic compounds, such as amines, in aerosol that originated in polar marine air masses (Dall'Osto et al., 2012). Gunsch et al. (2017, Supplement) briefly mentioned the detection of particulate TMA at a coastal Alaskan site in summer. However, our knowledge about the influence of amines on Arctic aerosol number concentration, size and chemical composition remains incomplete. Based on chamber studies of enhanced sulfuric acid nucleation rates due

to the presence of amines (Almeida et al., 2013), some studies have speculated that amines contribute to particle nucleation and

growth in the summertime Arctic (Leaitch et al., 2013; Croft et al., 2016b). For this reason the main objective of this research is to investigate emission sources and aerosol chemistry processes of particulate TMA in the summertime Arctic. We used aircraft-based single particle chemical composition measurements conducted in the Arctic summer. In addition, we analyze concurrent data from further aerosol and trace gas instruments as well as Lagrangian modeling simulations from FLEXPART.

This study provides an important opportunity to advance our understanding of the strong biological control over summertime Arctic aerosol.

## 2 Experimental and modeling section

### 2.1 Description of the sampling site and measurement platform

As one part of the NETCARE project (Network on Climate and Aerosols: Addressing Key Uncertainties in Remote Canadian

Environments), aircraft-based measurements were deployed from Resolute Bay, Nunavut (Canada) during 4 - 21 July 2014. In this study, we focus on measurements made during 4 - 12 July 2014. The satellite image from 4 July 2014 shown in Fig. 1 presents sea ice and open water conditions around Resolute Bay, which can be regarded as typical during 4 - 12 July. Six research flights (around 20 flight hours) were performed during this time. Flight tracks covered altitudes from 50 to 3000 m above continental as well as marine (partly covered with sea ice) regions (Fig. 2). Three flights aimed to sample above two

polynyas north of Resolute Bay. Notably, the sea ice south-east of Resolute Bay and close to the ice edge in Lancaster Sound was largely covered with melt ponds.

The instrument platform was the research aircraft Polar 6, a modified Basler BT-67 maintained by Kenn Borek and operated by the Alfred Wegener Institute for Polar and Marine Research (Herber et al., 2008). The aircraft was equipped with instruments to measure meteorological state parameters and several trace gases as well as aerosol particle number, size and chemical

composition. In general, aerosol instruments were connected to a forward-facing near-isokinetic stainless steel inlet, which was followed by a 1 inch-stainless steel manifold inside the cabin. All instruments were connected to the common inlet line system with 1/4-inch stainless steel tubing. Reactive trace gases were measured via a second PTFA inlet line. Further detailed information on the inlet and sampling strategy can be found in Leaitch et al. (2016), Willis et al. (2016), Burkart et al. (2017) and Aliabadi et al. (2016b).

### 2.2 Instrumentation

Number concentrations of particles greater than 5 nm in diameter ($N_{d>5nm}$) were measured with a TSI 3787 water-based ultrafine condensation particle counter (UCPC). Particle size distributions of particles greater than 250 nm ($N_{d>250nm}$) were measured using an optical particle counter from GRIMM (model 1.129 Sky-OPC). Measurements of carbon monoxide (CO) were

conducted with an Aerolaser ultra-fast CO monitor (model AL 5002). Sub-micron bulk aerosol composition was measured with an Aerodyne high-resolution time-of-flight aerosol mass spectrometer (HR-ToF-AMS). Operation of the HR-ToF-AMS

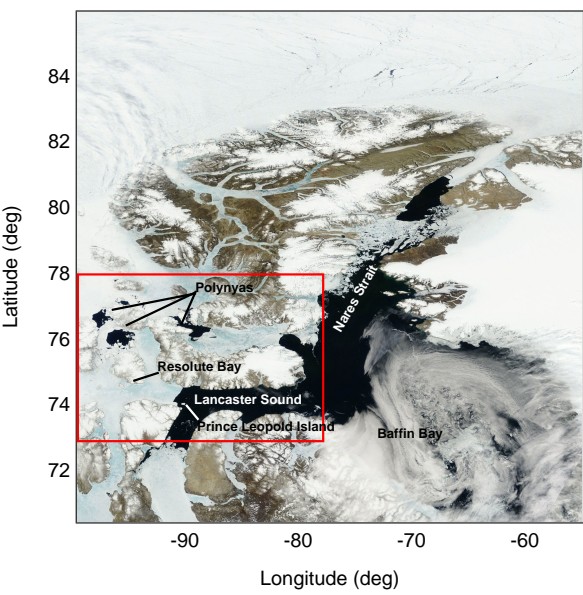

**Figure 1.** Satellite image (visible range from MODIS) from 4 July 2014 showing sea ice and open water conditions around Resolute Bay, in Lancaster Sound, Nares Strait and Baffin Bay. The red box indicates the region expanded with flight tracks in Fig. 2. Image is courtesy of NASA Worldview: https://worldview.earthdata.nasa.gov.

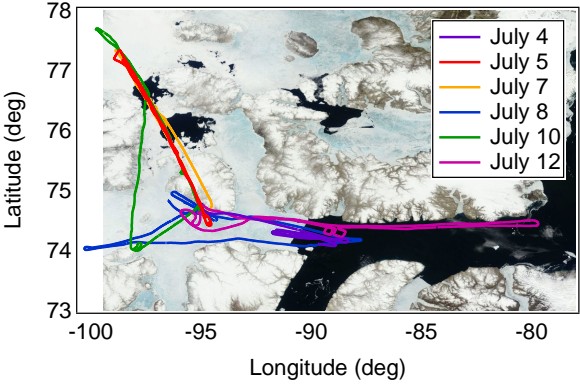

**Figure 2.** Satellite image (visible range from MODIS) from 4 July 2014 with a compilation of flight tracks conducted during 4 - 12 July 2014 (indicated with different colors).

aboard Polar 6 during NETCARE is described in Willis et al. (2016, 2017). State parameters and meteorological measurements were made using an AIMMS-20 from Aventech Research Inc. Detailed information on measurement principles and instrument calibrations are given in Leaitch et al. (2016) and Aliabadi et al. (2016b).

In order to provide information about the chemical composition of single aerosol particles, the ablation aerosol mass spec-

trometer ALABAMA (Aircraft-based Laser Ablation Aerosol Mass Spectrometer; Brands et al. (2011)) was deployed on the Polar 6 during NETCARE 2014. The basic measurement principle of the ALABAMA is as follows: first, the particles enter the system through a constant-pressure inlet. While ambient pressure changes, this device (custom-made at the Max Planck Institute for Chemistry) maintains a constant pressure in the following aerodynamic lens by varying the volume flow rate into the instrument. A flexible orifice is either squeezed or relaxed, depending on atmospheric pressure, by a bottom and top plate that are connected to a rotor. After passing through the inlet, particles are focused into a narrow beam with the help of a Liu-type aerodynamic lens (Liu et al., 1995a, b). The focused particles are detected by two light scattering signals (using 405 nm laserdiodes and photo-multipliers) allowing the determination of the size-dependent particle velocity. By comparing these values with the velocity of manufactured monodisperse polystyrene latex particles in five sizes ranging from 190 to 800 nm, we can derive the particle vacuum aerodynamic diameter ($d_{va}$). Next, the particles enter the ablation and ionization region in the high-vacuum system. The particles are ablated and ionized by a single triggered laser shot (266 nm, frequency-quadrupled Nd:YAG laser). In the final step, cations and anions produced by laser ablation are guided into a bipolar Z-ToF (Z-shaped Time of Flight) mass spectrometer, which provides bipolar mass spectra of individual particles. Due to limitations of the aerodynamic lens transmission efficiency and the lower detection limit of the photo-multipliers, the ALABAMA covers a particle size range from approximately 200 to 1000 nm.

## 2.3 FLEXPART- Lagrangian particle dispersion model

FLEXPART (FLEXible PARTicle dispersion model (here: version 10.0)) is a Lagrangian particle dispersion model (e.g., Stohl et al., 2005). For our analysis, we used operational analysis data from the European Centre for Medium-Range Weather Forecast (ECMWF) with 0.125° spatial and three hour time resolution. FLEXPART was operated in backward mode to provide potential emission sensitivity (PES) maps, which are the response functions to tracer releases from a receptor location. The value of the PES function is related to the particles' residence time in the output grid cell (for more details see Sect. 5 in Stohl et al. (2005) and Stohl (2006)). We used such PES maps together with sea ice and open ocean coverage derived from the satellite image in Fig. 1 to determine the total residence time of the measured air mass above open water regions three days prior to sampling in altitudes up to 340 m. The model output frequency was set up to one hour and 0.125° spatial resolution.

## 2.4 Single particle spectra analysis

In total, 7412 particles were chemically analyzed (mass spectra produced) by the ALABAMA during the study. 94 % of these spectra include size information. 80 % of these spectra have dual-polarity. Considering the 20 % single-polarity spectra, potential reasons for the lack of negative ions are discussed in the Supplement Sect. 1. Briefly, it is likely that single-polarity spectra are produced in high relative humidity (RH) environments (Neubauer et al., 1998; Spencer et al., 2008), in particular marine environments (Guasco et al., 2014).

The software package CRISP (Concise Retrieval of Information from Single Particles) was used to perform m/z (mass to ion charge ratio) calibration of particle mass spectra and peak area integration as well as to classify particle mass spectra using

ion markers for different species (Klimach, 2012). The marker method requires knowledge of certain ion markers belonging to a certain substance as well as knowledge of a certain marker threshold (ion peak area threshold). The typical fragmentation pattern of a substance due to laser ablation is crucial for defining the distinct ion markers. Fragmentation depends on laser wavelength and energy. Ion markers of many species are already well-known from laboratory measurements with the AL-

ABAMA (Schmidt et al., 2017) and additionally from literature of other single particle mass spectrometers (SPMS) using the same ablation laser wavelength. Table 1 lists ion markers of substances used in this study to identify the external and internal mixing state of particles. The identification of ion markers m/z +59 and +58 for TMA by Angelino et al. (2001) was confirmed by additional laboratory measurements with the ALABAMA (Supplement Sect. 2). To decide whether an ion signal is present, we used an ion peak area threshold of 10 mV and 25 mV for positive and negative mass spectra, respectively. Both thresholds

are chosen as a conservative measure on the basis of signal intensities of the non-occupied m/z values. Supplement Sect. 3 presents a detailed explanation of ion peak area threshold determination.

## 3 Results and Discussions

### 3.1 Meteorological conditions during the NETCARE 2014 campaign

The measurement period from 4 - 12 July 2014 was characterized by generally clear skies, calm wind speeds (Fig. 3) and

occasional scattered to broken stratocumulus clouds (Leaitch et al., 2016) due to prevailing high pressure influence in the Resolute Bay region. Based on low CO mixing ratios, low aerosol number concentrations (Fig. 3) and backward trajectory analysis, air masses measured in this period experienced a weak mid-latitudinal influence and were mainly affected by local emission sources (also denoted as "Arctic air mass period" in Burkart et al. (2017)). As shown in Fig. 2, our measurements took place largely over remote areas, which are dominated by Arctic vegetation, open water regions (e.g., polynyas, Lancaster

Sound) and sea ice coverage. Furthermore, seabird colonies were located close to the ice edge in Lancaster Sound and are likely a source of ammonia (Wentworth et al., 2016). Anthropogenic emissions might have affected our measurements, but are mainly related to the sparse Arctic settlements (Aliabadi et al., 2015) and can be ruled out by comparison with other tracers (e.g., CO). We can therefore expect that our observations from 4 - 12 July 2014 were mainly influenced by Arctic marine and terrestrial emissions.

As evident from vertical profiles of equivalent potential temperature (Theta$_e$) (Fig. 3), the mean upper boundary layer (BL) height for this measurement period was at around $340 \pm 100$ m. The vertical resolution of the profile in Fig. 3 (100 m) justifies the range of the mean BL height. The mean BL height within its range can be confirmed by results from an extensive study on BL height, mixing and stability during the NETCARE 2014 campaign (Aliabadi et al., 2016a). The capping temperature inversion above 390 m, inferred from values of Theta$_e$, represents a transport barrier for air masses between the BL and the

free troposphere (FT). The BL, compared to the FT, was characterized by lower wind speeds, higher RH and enhanced $N_{d>5nm}$ in contrast to $N_{d>250nm}$, indicating an enhanced number of ultrafine particles due to nucleation in the Arctic BL. A detailed discussion on this topic is given in Burkart et al. (2017).

**Table 1.** Marker species (with acronyms) and associated ion markers used in this study. Further given are references (SPMS lab and field studies) used for the assignment of ion markers as well as additional comments on marker species and ions.

| Marker species (Acronym) | Ion markers | References (lab/field studies) | Comments |
|---|---|---|---|
| Trimethylamine (TMA) | m/z +59 ($(CH_3)_3N^+$); +58 ($C_3H_8N^+$) | [1] / [2] [3] [4] | Additionally examined in laboratory measurements with ALABAMA (Supplement Sect. 2) |
| Sodium and chloride (Na/Cl) | m/z +23 ($Na^+$); (at least two of the following ions) +46 ($Na_2^+$), +62 ($NaO^+$), +63 ($NaOH^+$); (at least two of the following ions) +81/+83 ($Na_2Cl^+$), -35/-37 ($Cl^-$), -93/-95 ($NaCl_2^-$) | [5] [6] / [7] [2] | Sodium and chloride as indicators [5] [8] for sea spray particles<br><br>Isobaric interference with MSA at m/z -95 |
| Elemental carbon (EC) | (at least six of the following ions) m/z +36, +48, +60, ..., +144 ($C_{3-12}^+$) and/or (at least six of the following ions) m/z -36, -48, -60, ..., -144 ($C_{3-12}^-$) | [6] / [2] [9] | Except m/z -96 ($C_8^-$) due to the isobaric interference with $SO_4^-$ |
| Levoglucosan | (at least two of the following ions) m/z -45 ($CHO_2^-$), -59 ($C_2H_3O_2^-$), -71 ($C_3H_3O_2^-$) | [10] / [11] | Levoglucosan as indicator for biomass burning (BB) particles [12] |
| Potassium (K) | m/z +39 ($K^+$) | [6] / [2] [9] [13] [14] | K-dominant SPMS spectra associated with BB particles [10] [13] [14] [15] |
| Ammonium ($NH_4$) | m/z +18 ($NH_4^+$) | [9] / [2] [6] | |
| Methanesulfonic acid (MSA) | m/z -95 ($CH_3O_3S^-$) | [16] / [17] | Isobaric interference with $PO_4^-$ can be excluded due to missing ion signal for $PO_3^-$ at m/z -79 |
| Sulfate (S) | (at least one of the following ions) m/z -97 ($HSO_4^-$), -96 ($SO_4^-$) | [9] / [2] [6] | |

The semicolon (;) used in the list of ion markers serve as "and". Given reference numbers are defined as follows: [1] Angelino et al. (2001), [2] Roth et al. (2016), [3] Healy et al. (2015), [4] Rehbein et al. (2011), [5] Prather et al. (2013), [6] Schmidt et al. (2017), [7] Sierau et al. (2014), [8] O'Dowd and de Leeuw (2007), [9] Brands et al. (2011), [10] Silva et al. (1999), [11] Corbin et al. (2012), [12] Simoneit et al. (1999), [13] Hudson et al. (2004), [14] Pratt et al. (2011), [15] Pratt and Prather (2009), [16] Silva and Prather (2000), [17] Gaston et al. (2010).

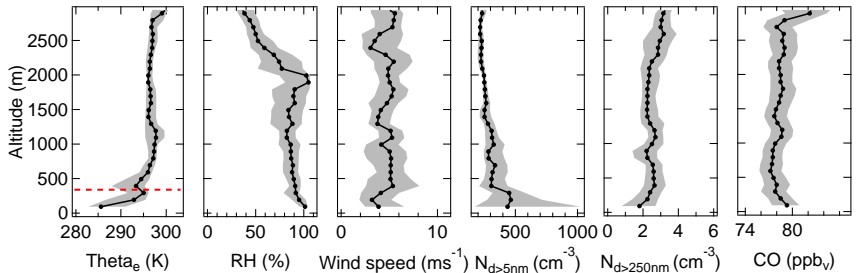

**Figure 3.** Vertically resolved median (black line) and interquartile ranges (gray shaded area) of the equivalent potential temperature (Theta$_e$), relative humidity (RH), wind speed, particle number concentration measured by the UCPC (N$_{d>5nm}$) and Sky-OPC (N$_{d>250nm}$) as well as CO mixing ratio (including all conducted flights from 4 - 12 July 2014). Measurements of N$_{d>250nm}$ started on 8 July due to prior technical issues. The red line depicts the derived mean upper height of the boundary layer during this measurement period (approximately 340m).

## 3.2 Size- and vertically resolved aerosol composition

Applying the marker method (Sect. 2.4), we classified 6676 particle mass spectra (90 % of the mass spectra analyzed by the ALABAMA (Sect. 2.4)) into five distinct particle types: TMA-, Na/Cl-, EC-, levoglucosan- and K-containing particles. TMA-, levoglucosan- and K-containing particles, with relative fractions of 23 %, 18 % and 46 %, respectively, appear to be the most
5 prominent particle types. Other alkylamines (other than TMA) and amino acids could not be identified (Supplement Sect. 4). Furthermore, only 2 % and 1 % of all analyzed particles are assigned as EC- and Na/Cl-containing particles, respectively. To obtain 100 % as total particle number, every spectrum is classified into one distinct particle type in the order presented in Tab. 2. The mean spectra in Fig. 4 combined with the additional ion signals listed in Tab. 2 provide an overview of the average chemical composition of each particle type. 28 % and 9 % of TMA- and K-containing particle spectra lack negative ions,
respectively. Potential reasons for the lack of negative ions are discussed in the Supplement Sect. 1. The mean spectrum of the remaining 736 particles (10 % of mass spectra analyzed by the ALABAMA), which could not be classified into one of the five particle groups outlined above, is shown in Fig. S7. For the further analysis we summarize these remaining particles in "others".

In order to describe the unique characteristics of TMA-containing particles compared to other particle groups, Fig. 5 and
15 6 depict the size and vertical distribution of each particle type, respectively. Both figures show the fractional abundance of each particle type per size and altitude bin, respectively. We show relative numbers of particles in order to eliminate the size-dependent transmission and detection efficiency of the ALABAMA (Fig. 5) and the dependence of the number of detected particles on sampling time in different altitudes (Fig. 6). The following use of the word *fraction* always refers to number fraction measured by the ALABAMA.

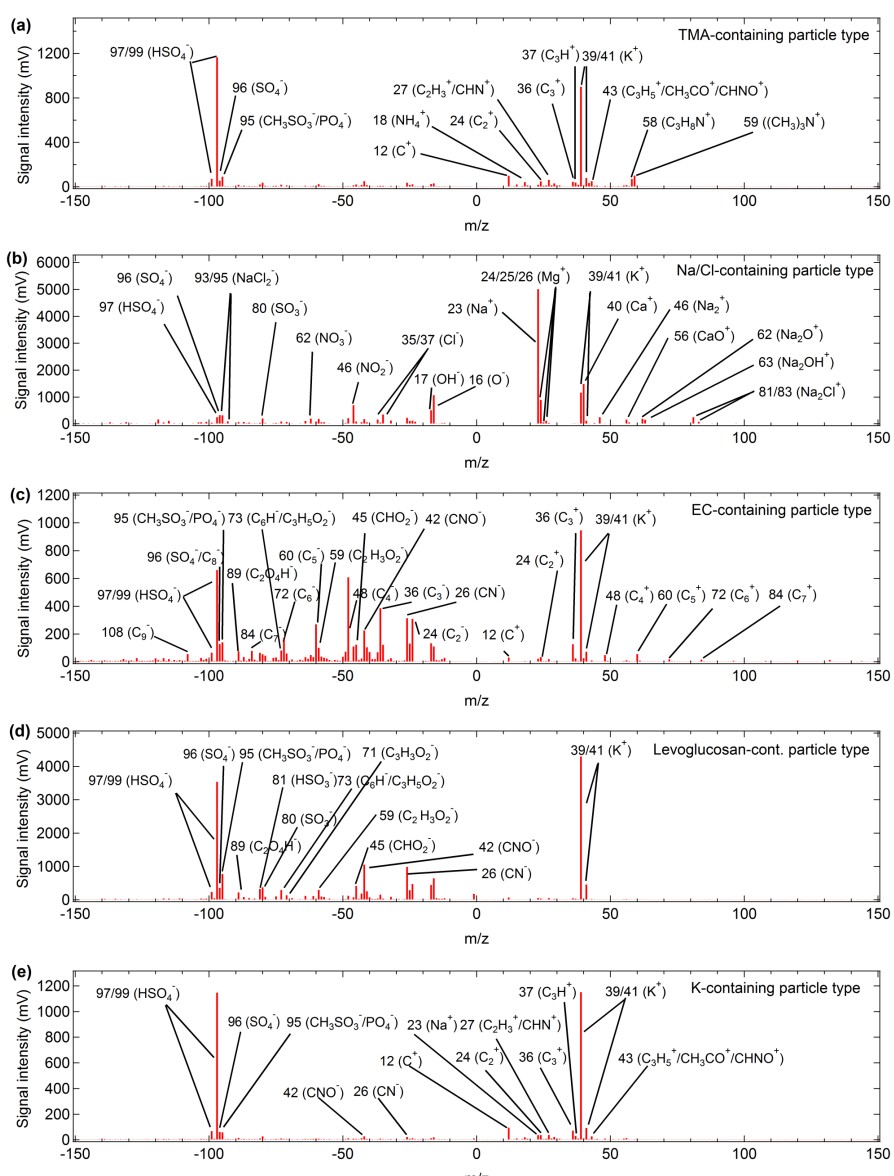

**Figure 4.** Bipolar mean spectra of the identified particle types: **(a)** TMA-containing (1688 particles ≙ 23 %), **(b)** Na/Cl-containing (106 particles ≙ 1 %), **(c)** EC-containing (138 particles ≙ 2 %), **(d)** levoglucosan-containing (1312 particles ≙ 18 %) and **(e)** K-containing (3432 particles ≙ 46 %).

### 3.2.1   Levoglucosan-, EC- and K-containing particle types

Levoglucosan, EC and potassium are known to be primarily produced from fossil fuel and biomass combustion processes (e.g., Bond et al., 2007; Simoneit, 2002; Andreae and Merlet, 2001; Simoneit et al., 1999). In particular, levoglucosan is formed

**Table 2.** Overview of the obtained five particle types and their internal mixing state derived from the mean spectra in Fig. 4 and 7. Additional ion signals of sulfate (m/z -97/99 ($HSO_4^-$), -96 ($SO_4^-$)) and potassium (m/z +39/41 ($K^+$)) were present in every mean spectrum and have therefore not been listed here. Further given are references (SPMS lab and field studies) used for the assignment of the additional ion signals to the corresponding chemical species.

| Particle type denotation | Characteristic ion signals in mean spectrum | Additional ion signals in mean spectrum | Corresponding chemical species |
|---|---|---|---|
| TMA-containing | m/z +59 ($N(CH_3)_3^+$), +58 ($NC_3H_8^+$) | m/z +18 ($NH_4^+$), $C_{1-3}^+$, +27 ($C_2H_3^+$/$CHN^+$), +37 ($C_3H^+$), +43 ($C_3H_5^+$/$CH_3CO^+$/$CHNO^+$) m/z -95 ($CH_3O_3S^-$) | ammonium carbon cluster ions hydrocarbons oxidized organics MSA |
| Na/Cl-containing | m/z +23 ($Na^+$), +46 ($Na_2^+$), +62 ($NaO^+$), +63 ($NaOH^+$), +81/+83 ($Na_2Cl^+$) m/z -35/-37 ($Cl^-$), -93/-95 ($NaCl_2^-$) | m/z +24/25/26 ($Mg^+$), +40 ($Ca^+$), +56 ($CaO^+$) m/z -26 ($CN^-$), -42 ($CNO^-$), -45 ($CHO_2^-$), -59 ($C_2H_3O_2^-$), -71 ($C_3H_3O_2^-$), -73 ($C_6H^-$/$C_3H_5O_2^-$), -46 ($NO_2^-$), -62 ($NO_3^-$) | magnesium[1][2] calcium[1] nitrogen-cont. organics[1][2][3][7] oxygen-cont. organics[3][4][5] nitrate[9] |
| EC-containing | $C_{1-7}^+$, $C_{1-8}^-$ | m/z -26 ($CN^-$), -42 ($CNO^-$), -45 ($CHO_2^-$), -59 ($C_2H_3O_2^-$), -73 ($C_6H^-$/$C_3H_5O_2^-$), -89 ($C_2O_4H^-$), -95 ($CH_3O_3S^-$) | nitrogen-cont. organics[3][6][7] oxygen-cont. organics[8][10][11] MSA |
| Levoglucosan-containing | m/z -45 ($CHO_2^-$), -59 ($C_2H_3O_2^-$), -71 ($C_3H_3O_2^-$) | m/z -26 ($CN^-$), -42 ($CNO^-$), -73 ($C_6H^-$/$C_3H_5O_2^-$), -89 ($C_2O_4H^-$), -95 ($CH_3O_3S^-$) | nitrogen-cont. organics[3][6][7] oxygen-cont. organics[8][10][11] MSA |
| K-containing | m/z +39/41 ($K^+$) | m/z +27 ($C_2H_3^+$/$CHN^+$), +37 ($C_3H^+$), +43 ($C_3H_5^+$/$CH_3CO^+$/$CHNO^+$) $C_{1-3}^+$, +23 ($Na^+$) m/z -26 ($CN^-$), -42 ($CNO^-$), -95 ($CH_3O_3S^-$) | hydrocarbons oxidized organics carbon cluster ions sodium nitrogen-cont. organics[3][6][7] MSA |

Given reference numbers are defined as follows: [1]Prather et al. (2013), [2]Guasco et al. (2014), [3] Pratt et al. (2009a), [4] Schmidt et al. (2017), [5]Trimborn et al. (2002), [6]Silva et al. (1999), [7]Fergenson et al. (2004), [8]Zauscher et al. (2013), [9]Brands et al. (2011), [10]Pratt et al. (2011), [11]Sullivan and Prather (2007).

via the breakdown of cellulose during biomass burning processes. The size distributions of levoglucosan- and EC-containing

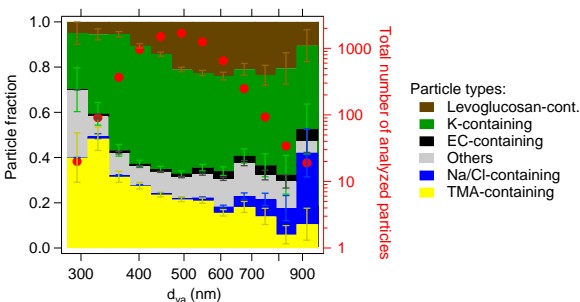

**Figure 5.** Cumulative size-resolved aerosol composition of the identified particle types (normalized to the total number of particles analyzed by the ALABAMA (indicated by red dots)): TMA-containing (yellow), Na/Cl-containing (blue), EC-containing (black), levoglucosan-containing (brown), K-containing (green) and "others" (gray). The errors associated with number fractions of the identified particle types were calculated using binomial statistics.

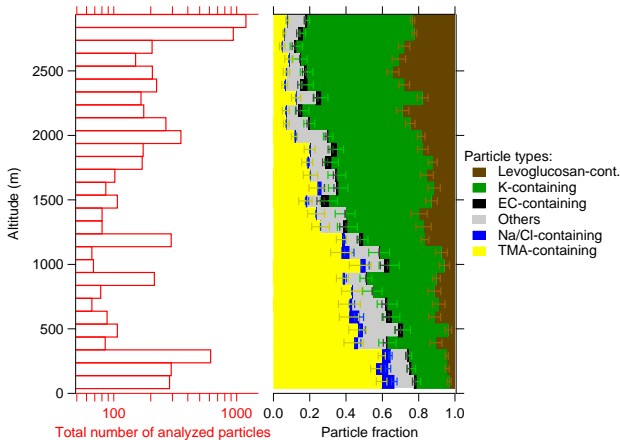

**Figure 6.** Cumulative vertically resolved aerosol composition of the identified particle types (normalized to the total number of particles analyzed by the ALABAMA (indicated by red bars)): TMA-containing (yellow), levoglucosan-containing (brown), Na/Cl-containing (blue), EC-containing (black), K-containing (green) and "others" (gray). There are in general two levels (below 340 m and above 2700 m) with enhanced number of particles analyzed by the ALABAMA, which is caused by a longer sampling time within these altitudes.

particles are shifted towards larger diameters compared to other particle types (Fig. 5). This result suggests these particles were exposed to chemical aging during long-range transport from biomass burning sources. K-containing particles are more evenly distributed across the size distribution (280 - 970 nm). EC-, levoglucosan- and K-containing particles contain mixtures of sulfate (m/z -97/99 ($HSO_4^-$)), MSA (m/z -95 ($CH_3SO_3^-$)) and organic nitrogen compounds (m/z -26 ($CN^-$), m/z -42 ($CNO^-$)) (Fig. 4c-e and Tab. 2). Further given that the $K^+$ ion signals (m/z +39/41) are dominant in mean cation spectra (Fig. 4c-e), we can likely attribute these particles to a biomass burning source (e.g., Silva et al., 1999; Hudson et al., 2004; Pratt and Prather, 2009; Pratt et al., 2011). Furthermore, Zauscher et al. (2013) assigned negative ion signals at m/z -73 ($C_3H_5O_2^-$)

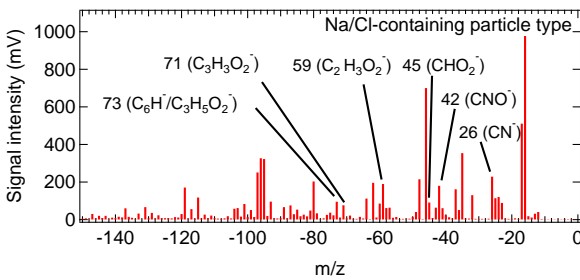

**Figure 7.** Expanded mean anion spectrum of 106 (1 %) Na/Cl-containing single particle spectra from Fig. 4b. Only the organics peaks are highlighted here.

to glyoxylic acid, which is typically present in biomass burning related SPMS spectra. Pratt et al. (2011) analyzed biomass burning particles internally mixed with oxalic acid (m/z -89 ($C_2O_4H^-$)). Both peaks are present in EC and levoglucosan mean mass spectra (Fig. 4c,d and Tab. 2). Previous Arctic SPMS studies by Sierau et al. (2014) and Gunsch et al. (2017) reported a particle type similar to our EC-containing particles (noted as ECOC type 1 and soot, respectively). Sierau et al.

(2014) attributed this particle type to remote biomasss/biofuel sources of continental origin. In contrast, Gunsch et al. (2017) assigned a large fraction of soot particles to emissions from the nearby oil fields at Prudhoe Bay. In the present study, the remote location of Resolute Bay excludes a larger influence of oil and gas extraction activities (Aliabadi et al., 2015; Peters et al., 2011). Further, Sierau et al. (2014) analyzed a particle type similar to the K-containing type in this study and noted as K-CN-sulfate type. They have speculated about a marine origin of these mixtures of potassium, sulfate and organic nitrogen

fragments. Sodium and MSA were partially present in the K-containing particle type in our study (Fig. 4e and Tab. 2), which confirms the hypothesis of Sierau et al. (2014). However, it is likely that this large group of K-containing particles (46 %) includes different emission sources within and above the local BL.

The vertical dependence in EC-containing particles is not further analyzed here due to the low statistical significance of 138 particles detected over the entire study at all altitudes. From the vertical profile of levoglucosan- and K-containing particles

given in Fig. 6, it can be seen that their fractions increase with increasing altitude. These observations correspond to enhanced CO mixing ratios and $N_{d>250nm}$ (Fig. 3) providing further evidence for biomass burning as the source of levoglucosan- and K-containing particles. Despite the potential for oxidation of levoglucosan during transport, it has been previously reported as associated with biomass burning aerosol in Arctic regions (Hu et al., 2013; Fu et al., 2013, 2009). Sierau et al. (2014) and Gunsch et al. (2017) did not report the detection of levoglucosan with SPMS measurements in the summertime Arctic. It is

likely that these ground-based measurements missed a large fraction of particles typically present above the BL (including levoglucosan particles).

### 3.2.2 Na/Cl-containing particle type

A number of studies have reported on the primary production of sea spray particles via bubble bursting at the sea surface (e.g., Blanchard and Woodcock, 1980; O'Dowd and de Leeuw, 2007). Na/Cl-containing particles observed in this study show particle diameters mainly larger than 600 nm and they primarily exist at lowest altitudes. Thus, the Na/Cl-containing particle type can be associated to locally emitted sea spray. The occurrence of sulfate and nitrate ion signals in the mean spectrum (Fig. 4b and Tab. 2) suggests that some particles have already been exposed to chemical aging via reactions with sulfuric and nitric acid forming nitrate and sulfate and releasing HCl to the gas phase (e.g., Gard et al., 1998; O'Dowd et al., 1999; Sorensen et al., 2005). Similar ion peaks were observed by Sierau et al. (2014) and Gunsch et al. (2017) and assigned to aged sea spray particles. Internal mixing of Na/Cl-containing particles with MSA cannot be finally ruled out since $NaCl_2^-$ and MSA have an isobaric interference at m/z -95 (Tab. 1). However, due to the concurrent existence of other Na and Cl ion signals as well as signals at m/z -93 (isotope of $NaCl_2^-$), it is likely that ion signals at m/z -95 are largely produced by $NaCl_2^-$.

Interestingly, some of the Na/Cl-containing particles are internally mixed with different inorganics (such as magnesium and calcium) as well as oxygen- and nitrogen-containing organic compounds, as indicated by the mean spectrum in Fig. 4b and Fig. 7. It is known from previous SPMS laboratory studies on sea spray particles produced from biologically active waters that organic nitrogen species present on inorganic salts arise from biological activity (Prather et al., 2013; Guasco et al., 2014). In particular, organic nitrogen fragments together with calcium, sodium and phosphate have been linked to signatures of biological species (e.g., Pratt et al., 2009a; Schmidt et al., 2017). SPMS spectra of biological particles presented in Pratt et al. (2009a) further indicate the occurrence of oxygen-containing organic compounds at m/z -71 ($C_3H_3O_2^-$). Laboratory studies with the ALABAMA investigating biological species (such as bacteria and pollen) also showed the existence of negative ion signals at m/z -45 ($C_2H_5O^-$), m/z -59 ($C_3H_7O_2^-/C_3H_9N^-$) and m/z -71 ($C_3H_3O_2^-/C_4H_7O^-$) in addition to the presence of phosphate and organic nitrogen compounds (Schmidt et al., 2017). Anion signals at m/z -26 ($C_2H_2^-$) and m/z -42 ($C_2H_2O^-/C_3H_6^-$) can be further attributed to cellulose (Schmidt et al., 2017). Moreover, Trimborn et al. (2002) reported the concurrent presence of sodium, chloride and oxygen-containing organic compounds (m/z -73 ($C_3H_5O_2^-$) and m/z -59 ($C_2H_3O_2^-$)) in ambient SPMS spectra and attributed them to organic containing sea salt particles. Other Non-SPMS studies (e.g., X-ray microscopy methods) have reported the occurrence of organic-rich (e.g., carboxylate) sea spray particles originating from microorganisms and organic compounds enriched in the sea surface microlayer in mid-latitude oceans (e.g., Quinn et al., 2014; Blanchard and Woodcock, 1980) and in Arctic regions (e.g., Wilson et al., 2015; Frossard et al., 2014; Hawkins and Russell, 2010; Russell et al., 2010). Taken together, the presence of magnesium and calcium together with nitrogen- and oxygen-containing organic species in sea spray particles suggests that such organic fragments have a marine-biogenic origin.

### 3.2.3 TMA-containing particle type

TMA-containing particles have several characteristics that are contrary to the other particles types. The size distribution of TMA-containing particles is shifted towards smaller diameters (Fig. 5) and the fractional abundance increases with decreasing altitude (Fig. 6). In addition, TMA-containing particles detected within the BL are smaller compared to particles observed

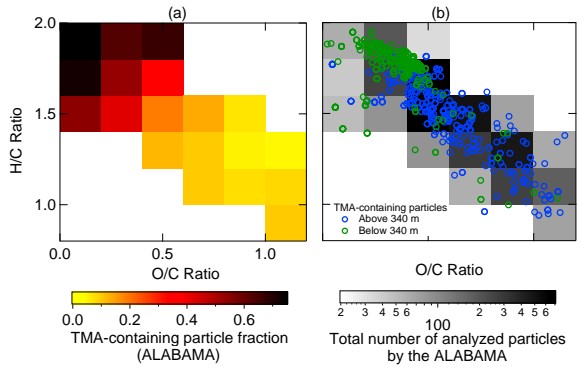

**Figure 8.** Comparison between the HR-ToF-AMS estimated oxygen-to-carbon (O/C) and hydrogen-to-carbon (H/C) ratios colored by: **(a)** TMA-containing particle number fraction (ALABAMA) and **(b)** total number of analyzed particles by the ALABAMA (gray to black) as well as the presence of particulate TMA above (blue circles) and below (green circles) 340 m (mean upper BL height, Sect. 3.1).

aloft (Fig. S8). Comparison of HR-ToF-AMS estimated oxygen-to-carbon (O/C) and hydrogen-to-carbon (H/C) ratios with the ALABAMA particulate TMA fraction gives an indication of the degree of particle oxidative aging (e.g., Jimenez et al., 2009; Heald et al., 2010; Ng et al., 2011; Willis et al., 2017). Less oxygenated organics measured with the HR-ToF-AMS were present when the fraction of TMA-containing particles was high (Fig. 8a, up to 75 % in the upper left corner). This suggests

that a large fraction of particulate TMA, especially within the BL (indicated with green circles in Fig. 8b), had not been subject to extensive oxidative aging. According to these results together with the existence of a stable stratified BL (Fig. 3), we can infer that particulate TMA present within the Arctic BL originated from inner-Arctic sources. Possible inner-Arctic sources of TMA, referring to Ge et al. (2011a), are oceanic phytoplankton biomass or other marine organisms and various human activities (e.g., waste incineration, vehicle exhaust, residential heating). Gaseous TMA emissions may then take part in aerosol

chemistry in several ways including acid-base reactions, oxidation processes, dissolution in cloud droplets and nucleation (e.g., Ge et al., 2011a, b; Rehbein et al., 2011; Erupe et al., 2011; Murphy et al., 2007; Angelino et al., 2001).

The mean spectrum of TMA-containing particles (Fig. 4a) shows no indications that further N-containing compounds (such as amine oxidation products, e.g., amides, nitramines and imines) other than TMA (with specific ion signals at m/z +59 and +58) were present on these particles. Figure 4a and Table 2 further illustrate an internal mixing of sulfate and TMA, which

indicates that aminium sulfate salts may be present (e.g., Murphy et al., 2007; Barsanti et al., 2009; Smith et al., 2010). We can therefore hypothesize that the formation of particulate TMA was accompanied with acid-base reactions including TMA, sulfuric and methanesulfonic acid (e.g., Facchini et al., 2008). Rehbein et al. (2011) reported enhanced gas-to-particle partitioning of TMA by dissolution in cloud/fog droplets and subsequent formation of aminium salts. Thus, it is further possible, due to the occasional presence of low-level clouds (Leaitch et al., 2016), that the formation of TMA-containing particles was

favored by pre-existing wet and acidic particles.

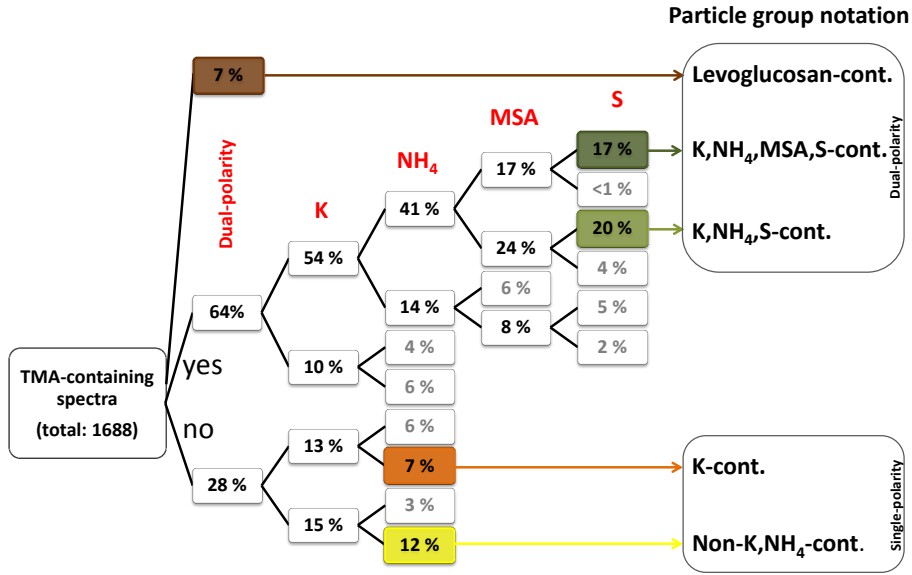

**Figure 9.** Classification of TMA-containing particles on basis of their different internal mixing states. Each branch describes the existence or non-existence of several substances (potassium (K), ammonium ($NH_4$), MSA and sulfate) on TMA-containing particles with relative abundances normalized to the occurrence of 1688 TMA-containing particles. An initial query regarding the existence of dual-polarity spectra is included. Based on this classification four TMA-containing particle sub-types arise (colored boxes with relative fractions): "K,$NH_4$,MSA,S-", "K,$NH_4$,S-", "K-" and "Non-K,$NH_4$-containing". We further considered an internal mixing of particulate TMA with levoglucosan (7 %), Na/Cl and EC (not listed here), whereby the latter two types with relative fractions of less than 1 % are negligible for the further analysis. Gray-shaded numbers indicate groups with relative fractions of less than 7 % that are not further considered.

### 3.3 Internal mixing state of TMA-containing particles

The internal mixing state of TMA-containing particles was further classified by applying the marker method introduced in Sect. 2.4 and Tab. 1 for compounds that are apparent in the mean spectrum (Fig. 4a and Tab. 2), such as potassium (K), ammonium ($NH_4$), MSA and sulfate (S). The diagram in Fig. 9 illustrates the classification algorithm as follows: an upper branch always refers to a positive response ("yes") for whether different ion markers are present in spectra or not; a lower branch shows the opposite answer ("no"). Besides the substances that already appeared in the mean spectrum of TMA-containing particles here TMA-containing spectra are also viewed based on the concurrent existence of levoglucosan, Na/Cl and EC. We did not consider in detail the concurrent existence of carbon cluster ions (m/z +12, +24, ..), different hydrocarbons (m/z +27 and +37) and oxidized organics (m/z +43) since 90 % of all TMA-containing particles contain at least one of these ion signals. The classification of the TMA-containing particle type is further based on an initial differentiation between dual- and single-polarity mass spectra. As can be seen in Fig. 9, 28 % of TMA-containing particle spectra lack negative ions. Consequently, we cannot state if

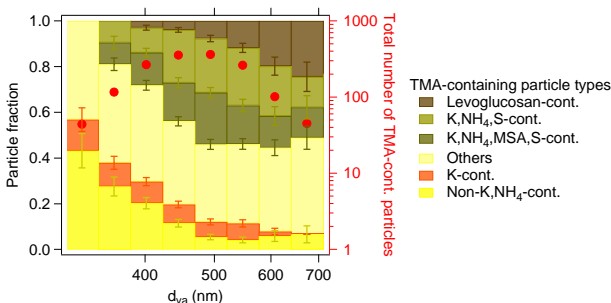

**Figure 10.** Cumulative size-resolved aerosol composition of TMA-containing particle sub-types (normalized to the total number of TMA-containing particles (indicated by red dots)): "K,NH$_4$,MSA,S-containing" (dark green), "K,NH$_4$,S-containing" (light green), "K-containing" (orange), "Non-K,NH$_4$-containing" (yellow), "levoglucosan-containing" (brown) and "others" (light yellow).

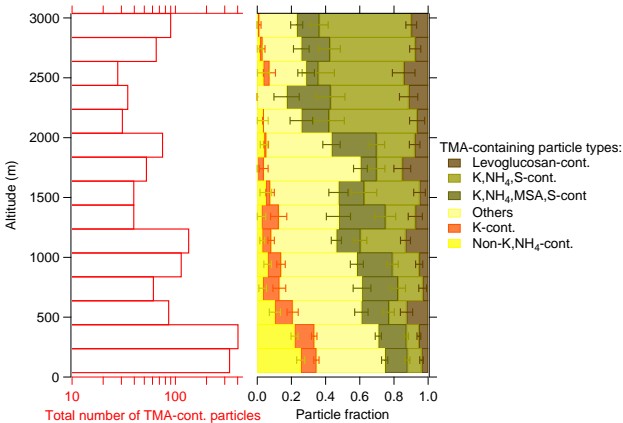

**Figure 11.** Cumulative vertically resolved aerosol composition of TMA-containing particle sub-types (normalized to the total number of TMA-containing particles (indicated by red bars)): "K,NH$_4$,MSA,S-containing" (dark green), "K,NH$_4$,S-containing" (light green), "K-containing" (orange), "Non-K,NH$_4$-containing" (yellow), "levoglucosan-containing" (brown) and "others" (light yellow).

species producing anions (such as MSA and sulfate) were present on these particles. Potential reasons for the lack of negative ions are discussed in the Supplement Sect. 1. Particle sub-group notation is based on the existence or non-existence of different species in TMA-containing particles. For reasons of clarity, particle types with less than 7 % fractional abundance (corresponding to a total number of less than 118 particles) are not explicitly considered in this analysis, but are summarized as "others".

5   Following the categorization in Fig. 9, five groups of different internal mixing states arise: "K,NH$_4$,MSA,S-", "K,NH$_4$,S-", "K-", "Non-K,NH$_4$-" and "levoglucosan-containing" particles. These five TMA particle sub-types will be divided into those containing biomass burning tracers (such as levoglucosan and potassium) and those not containing these tracers.

As can be seen in Fig. 9, a large fraction of TMA-containing particles (74 %) are additionally composed of biomass burning tracers such as potassium (67 %) and levoglucosan (7 %). According to Pöhlker et al. (2012), this internal mixture can be

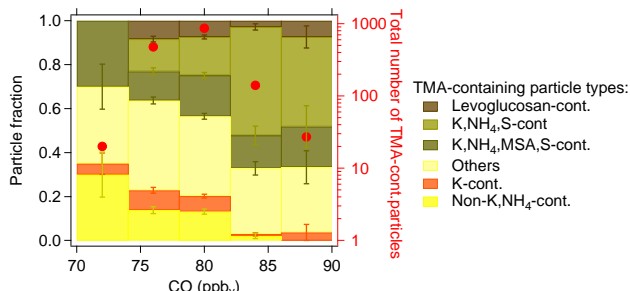

**Figure 12.** CO measurements compared with cumulative fraction of TMA-containing particle sub-types (normalized to all TMA-containing particles (indicated by red dots)): "K,NH$_4$,MSA,S-containing" (dark green), "K,NH$_4$,S-containing" (light green), "K-containing" (orange), "Non-K,NH$_4$-containing" (yellow), "levoglucosan-containing" (brown) and "others" (light yellow).

explained by potassium-containing particles acting as seeds for the condensation of organic material. Thus, the measured particulate TMA can be considered a secondary component that condensed on pre-existing primary particles. It is also conceivable that TMA particles containing potassium and levoglucosan are a result of biomass burning emissions (Schade and Crutzen, 1995; Ge et al., 2011a; Silva et al., 1999; Hudson et al., 2004; Pratt and Prather, 2009; Pratt et al., 2011). The size distribution

of the TMA particles containing levoglucosan is shifted towards larger diameters compared to other TMA particle sub-types. (Fig. 10). Moreover, Fig. 11 demonstrates that TMA particle sub-types including potassium and levoglucosan were more abundant above the BL in contrast to "Non-K,NH$_4$-containing" TMA particles. Comparison between CO mixing ratios and TMA sub-types abundance (Fig. 12) shows larger fractions of "K,NH$_4$,S-containing" and "levoglucosan-containing" TMA particle sub-types in higher CO environments compared to "Non-K,NH$_4$-containing" TMA particles. Taken together, these results

suggest that TMA particles containing levoglucosan and potassium likely originated from remote biomass burning emission sources and were transported to our measurement site.

Another large fraction (25 %, Fig. 9) of particulate TMA is neither internally mixed with potassium nor with any other tracer of biomass burning. This result suggests that these TMA-containing particles resulted from SOA formation. This is consistent with results from particle size distributions of TMA sub-types in Fig. 10 illustrating that the fractional abundance of "Non-

K,NH$_4$-containing" TMA particles is highest between 280 and 380 nm compared to other sub-types containing levoglucosan and/or potassium. In particular, positive ion mass spectra of the sub-type "Non-K,NH$_4$-containing" (12 % single-polarity (yellow box in Fig. 9) and 6 % dual-polarity (not colored in Fig. 9)) show ion signals only for carbon cluster ions and fragments of hydrocarbons (Fig. 13a,b). Due to a suppression of anion signals, likely in high RH environments (Supplement Sect. 1), we cannot state whether sulfate or MSA were present in these particles. However, the dual-polarity mean spectrum of the 6 %

TMA-containing particles not including potassium and ammonium (Fig. 13b, not colored in Fig. 9) indicates the concurrent presence of sulfate or MSA. From the absence of ammonium in these TMA particles containing sulfate or MSA, we can further conclude that aminium salts were present. This result demonstrates that amines, in addition to ammonia, may take part in the neutralization of acidic aerosol. This is of particular interest considering the reduced sources of ammonia in the Arctic and the

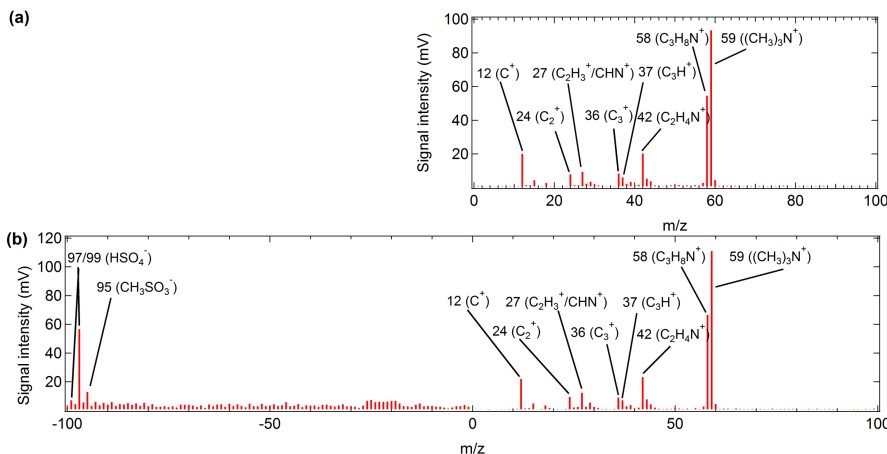

**Figure 13.** Mean spectra of "Non-K,NH$_4$-containing" TMA particle sub-type: **(a)** Single-polarity particle mass spectrum (12 %, yellow box in Fig. 9), **(b)** Dual-polarity particle mass spectrum (6 %, not colored in Fig. 9).

ocean as a net sink of NH$_3$ in the summertime Canadian Arctic (Wentworth et al., 2016). Furthermore, Fig. 14 indicates a positive correlation between MSA mass concentrations measured with HR-ToF-AMS and the fraction of "Non-K,NH$_4$-containing" TMA particles. Given that MSA can be used as an indicator for marine influence on sub-micron aerosol, we can conclude that the existence of an inner-Arctic marine-biogenic source of TMA is likely. Moreover, "Non-K,NH$_4$-containing" TMA particles

are most abundant at the lowest altitudes (Fig. 11) and are coincident with the presence of less aged particulate organic aerosol (Fig. 8). Taken together, the characteristics of the "Non-K,NH$_4$-containing" TMA particle sub-type suggest that gaseous TMA emissions from inner-Arctic sources (likely marine-biogenic) act as precursor for the formation of SOA within the summertime Arctic BL.

## 3.4   Source apportionment analysis of TMA-containing particles

This section will further explore potential emission sources of TMA in the Arctic BL. Thus, the following analysis was restricted to measurements below 340 m (mean upper BL height, Sect. 3.1). Figure 15 shows the temporal distribution of non-TMA-containing particles (such as Na/Cl-, EC- , levoglucosan- and K-containing) and TMA-containing sub-types. Figure 16 depicts the spatially resolved fraction of TMA-containing particles below 340 m (left panel) as well as the measured wind direction (right panel) for measurements on 4, 7 and 8 July. We further used three-day FLEXPART backward simulations (Sect.

2.3) for air mass history analysis of the three measurement legs (Fig. 17) to understand the source regions of TMA-containing

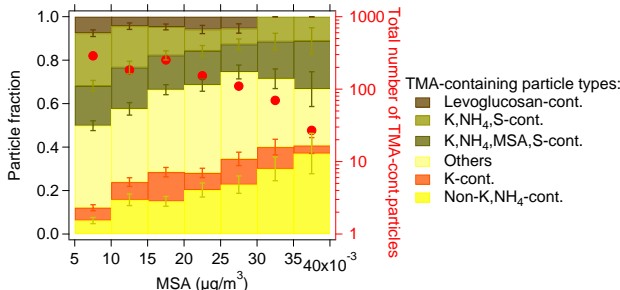

**Figure 14.** MSA concentrations measured with the HR-ToF-AMS compared with cumulative fraction of TMA-containing particle sub-types (normalized to all TMA-containing particles (indicated by red dots)): "K,NH$_4$,MSA,S-containing" (dark green), "K,NH$_4$,S-containing" (light green), "K-containing" (orange), "Non-K,NH$_4$-containing" (yellow), "levoglucosan-containing" (brown) and "others" (light yellow).

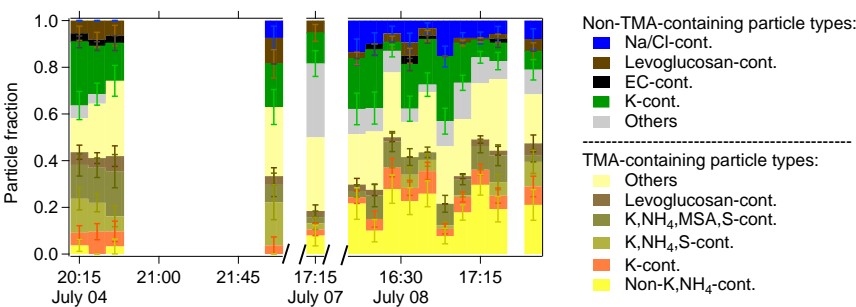

**Figure 15.** Temporally resolved aerosol composition of the identified non-TMA-containing particle types (normalized to the total number of particles analyzed by the ALABAMA): Na/Cl-cont. (blue), EC-cont. (black), levoglucosan-cont. (brown), K-cont. (green) and "others" (gray) as well as TMA-containing particle sub-types (normalized to the total number of particles analyzed by the ALABAMA): "levoglucosan-cont." (light brown), "K,NH$_4$,MSA,S-cont." (dark green), "K,NH$_4$,S-cont." (light green), "K-cont." (orange), "Non-K,NH$_4$-cont." (yellow) and "others" (light yellow). Fractional abundances of the particle types were calculated for 10 minute time intervals. Only time intervals with at least 20 measured particles were considered. Measurements within the BL on 5, 10 and 12 July did not provide any 10 minute time interval with more than 20 spectra.

particles.

Potential emission sensitivity (PES) maps combined with sea ice coverage (Fig. 1) show that air masses measured on 4 and 7 July spent less than one hour and around seven hours, respectively, in the previous three days in regions of open water (polynyas north of Resolute Bay and Nares Strait). On both days the air was mainly advected above sea ice and snow covered regions north of Resolute Bay (Fig. 17 compared with Fig. 1). The prevailing wind direction on 4 and 7 July along the flight tracks (Fig. 16) is from the north and east and therefore consistent with FLEXPART backward simulations (Fig. 17). From measurements on 4 and 7 July it is not possible to attribute TMA emissions to marine-biogenic or anthropogenic sources (e.g., vehicle exhaust, residential heating and waste incineration emissions in Resolute Bay). A more detailed air mass history analysis was

carried out on observations from 8 July.

The case of 8 July provides further evidence for a marine-biogenic influence on TMA-containing particles through secondary processes. The prevailing wind direction along the presented flight leg is from the east (Fig. 16) with low wind speeds up to a maximum of 7 m/s (Fig. S9). The fraction of TMA-containing particles decreases with a shift to a more southerly wind direction (yellow to green colors, Fig. 16). The highest fractional abundance of particulate TMA was measured close to the ice edge (Fig. 16) at low wind speeds close to zero m/s (Fig. S9). Thus, the ice edge in the western section of Lancaster Sound where the highest surface phytoplankton production rate and chlorophyll $a$ concentration were measured (M. Gosselin, personal communication) and large bird colonies at Prince Leopold Island (Fig. 1) (Wentworth et al., 2016) likely contribute to TMA emissions in the area. Consistent with these observations, previous aerosol chemical composition measurements on Bird Island in the South Atlantic (> 50°S) have reported the presence of amines and amino acids emitted from local fauna including seabirds, penguins and fur seals (Schmale et al., 2013). Further, air mass history predicted by FLEXPART three-day backward simulations (Fig. 17) illustrates that these air masses were advected at low levels above open water regions in Lancaster Sound, Baffin Bay and Nares Strait (compare with Fig. 1). Air masses measured during this flight leg on 8 July resided for more than 17 hours during the three days prior to sampling above regions of open water. Further, anthropogenic influences on amine emissions from nearby Resolute Bay are likely negligible since CO concentrations are very low. Another important finding is that primary sea spray particles (Na/Cl-containing) and TMA-containing particles measured on 8 July are externally mixed (Fig. 15) although both substances seems to be released from the ocean. This analysis solidifies the earlier hypothesis (Sect. 3.3) that particulate TMA presents secondary aerosol (Facchini et al., 2008). The higher abundance of the TMA-containing particle sub-type "Non-K,NH$_4$-containing" on 8 July (Fig. 15), compared to other days, further supports the hypothesis of SOA formation. It is further relevant to discuss that on 8 July from 15:50 UTC until 17:20 UTC (respective flight leg in Fig. 15) we flew low over sea ice in the vicinity of dissipating low-level clouds. These clouds had formed above the open water regions east of our flight leg (Burkart et al., 2017; Leaitch et al., 2016). We can therefore assume that cloud processing likely contributed to enhanced gas-to-particle partitioning of TMA as earlier reported in Rehbein et al. (2011). In addition, high organic-to-sulfate and MSA-to-sulfate ratios measured with the HR-ToF-AMS during this flight leg (see Sect. 4.3 in Burkart et al. (2017)) indicate that particle growth was driven by ocean-derived precursor gases (dimethylsulfide and organic species). Taken together, results from 8 July demonstrate secondary organic aerosol formation from marine-biogenic sources of gas-phase precursors, including TMA.

# 4   Conclusions

We presented results from aircraft-based single particle aerosol measurements in the summertime Canadian High Arctic. Our study has shown the presence of particulate TMA in the Arctic summer, comprising 23 % of all particles analyzed by the ALABAMA. SPMS measurements do not provide bulk analysis of aerosol chemical composition, therefore we can not obtain TMA mass concentrations. Nevertheless, the number of particles analyzed by the ALABAMA (> 7000) is sufficient to conduct a statistical analysis. This allows us to draw conclusions about mixing state, vertical and size distributions as well as potential

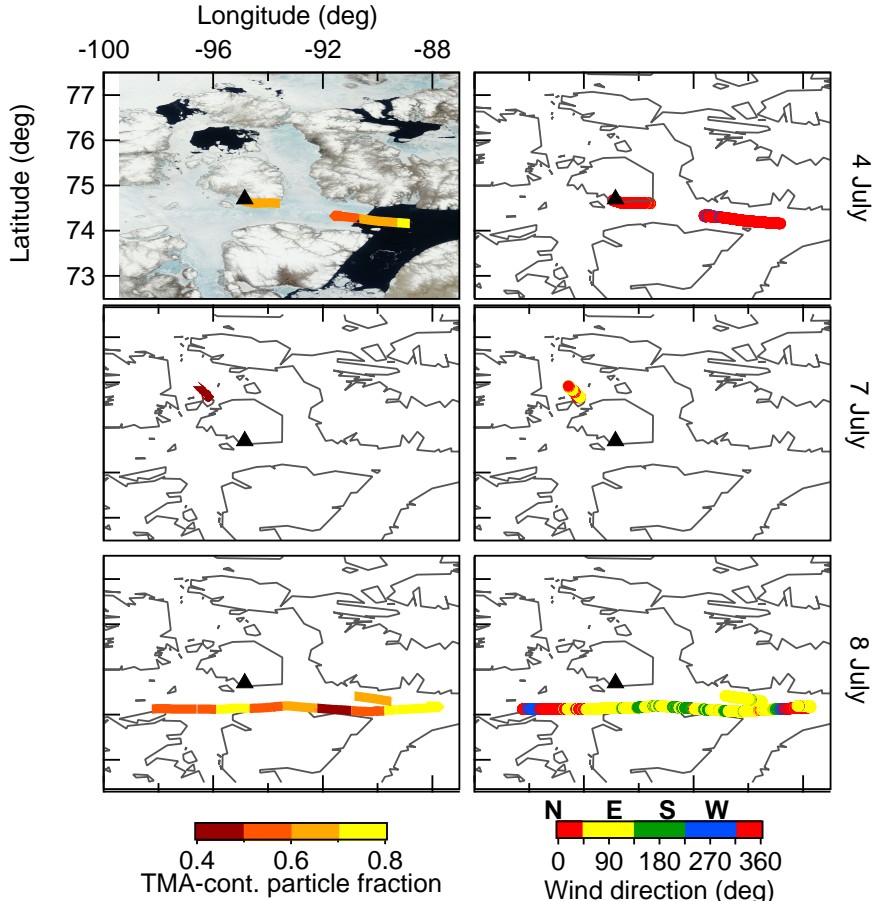

**Figure 16.** Spatially resolved fraction of TMA-containing particles (left column, color-coded) and wind direction (right column, color-coded) below 340 m. Different rows present different measurement days. The first graph additionally shows the satellite image on 4 July in the visible range. Further satellite images are not presented here due to negligible changes in sea ice coverage from 4 - 8 July. Abbreviations N, E, S and W refer to North, East, South and West. The black triangle presents the location of Resolute Bay on the map.

emission sources of particulate TMA in summertime Arctic regions.

We present two main sources of particulate TMA in the summertime Arctic. First, we show the presence of inner-Arctic marine-biogenic sources resulting in secondary aerosol formation by TMA, sulfate, MSA, ammonia and other organics. Second, we have indications for long-range transport from biomass burning sources. We measured the maximum occurrence of particulate TMA (approximately 60 %) in a clean and stable stratified Arctic BL. In addition, TMA-containing particles present within the Arctic BL were smaller and were associated with less aged organic aerosol compared to aerosol observed aloft. High fractions of particulate TMA were measured at low wind speeds (near zero m/s) and close to the biologically active marginal ice zone. Further, BL air masses including high fractions of particulate TMA spent a long time (more than 17 hours) prior to sampling

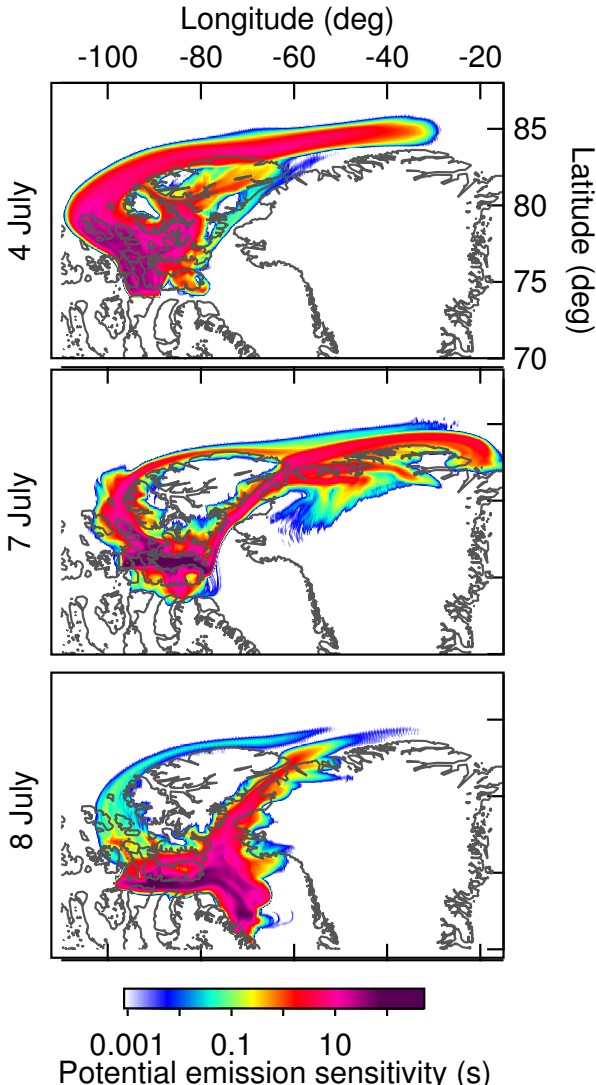

**Figure 17.** FLEXPART backward simulations of the considered measurement periods (Fig. 15) three days prior to sampling and at altitudes below 340 m. The color-coded area presents values of the potential emission sensitivity (PES) function in a particular grid cell (Sect. 2.3). Different rows depict different measurement days.

above Arctic open water regions. Moreover, the TMA particle sub-type containing MSA, sulfate and other organic species was more abundant when MSA mass concentrations (measured with HR-ToF-AMS) were high. Furthermore, the concurrent existence of sulfate, MSA and TMA in single particle spectra indicates the presence of aminium salts. This demonstrates that TMA may take part in neutralizing acidic aerosol along with ammonia. We additionally found that primary sea spray particles and TMA-containing particles are externally mixed although both substances are released by the ocean. It is further possible

that gas-to-particle partitioning of TMA was enhanced in the vicinity of clouds and fog through dissolution of TMA in droplets and subsequent acid-base reactions (Rehbein et al., 2011). In contrast to the marine inner-Arctic sources, we have evidence for particulate TMA from long-rang transport of biomass burning aerosol. We demonstrate that levoglucosan and potassium (biomass burning tracers) are internally and externally mixed with particulate TMA. These particle types were more abundant

above the Arctic BL as well as larger in particle sizes compared to particles not including these components.

Taken together, these findings contribute to our knowledge of marine-biogenic influences on secondary aerosol chemical composition and particle growth in the summertime Canadian Arctic. This is the first study demonstrating the incorporation of amines in Arctic aerosol from inner-Arctic sources. Based on spatial and temporal limitations of our measurements, it is difficult to assess how representative our findings are of the broader Arctic region. However, recent measurements confirm the

presence of particulate amines and its marine-biogenic source at another Arctic site (Alert, 82.5 °N) (Leaitch et al., 2017). Future wide-spread and long-term Arctic measurements of atmospheric amines would help to extend our results to other regions.

*Author contributions.* Jon Abbatt, Richard Leaitch and Andreas Herber designed the research project. Franziska Köllner, Johannes Schneider, Heiko Bozem, Richard Leaitch, Megan Willis and Julia Burkart carried out the measurements. Amir Aliabadi processed the wind measurements. Thomas Klimach, Frank Helleis and Johannes Schneider re-designed and further developed the ALABAMA for aircraft-

based measurements. Franziska Köllner analyzed the data with the help of Johannes Schneider, Peter Hoor, Thomas Klimach and Daniel Kunkel. Franziska Köllner wrote the manuscript. All co-authors commented on the manuscript.

*Acknowledgements.* The authors thank Kenn Borek Air Ltd., in particular our pilots Kevin Elke and John Bayes, as well as our aircraft maintenance engineer Kevin Riehl. We thank Jim Hodgson and Lake Central Air Services in Muskoka, Jim Watson (Scale Modelbuilders, Inc.), Julia Binder and Martin Gehrmann (Alfred Wegener Institute, AWI) for their support of the integration of the instrumentation in the

aircraft. We thank Bob Christensen (University of Toronto), Lukas Kandora, Manuel Sellmann, Christian Konrad and Jens Herrmann (AWI), Desiree Toom, Sangeeta Sharma (ECCC), Kathy Law and Jenny Thomas (LATMOS) for their support before and during the study. We thank Christiane Schulz (MPIC) for her support during the integration of the instruments in Muskoka. We thank the Biogeochemistry department of MPIC for providing the CO instrument and Dieter Scharffe for his support during the preparation phase of the campaign. We thank the Nunavut Research Institute and the Nunavut Impact Review Board for licensing the study. Logistical support in Resolute Bay was provided

by the Polar Continental Shelf Project (PCSP) of Natural Resources Canada under PCSP field project 218-14. Funding for this work was provided by the Natural Sciences and Engineering Research Council of Canada through the NETCARE project of the Climate Change and Atmospheric Research Program, the Alfred Wegener Institute, Environment and Climate Change Canada and the Max Planck Society. Special thanks to the whole NETCARE-team for data exchange, discussions and support.

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
