# Peer review of "Particulate trimethylamine in the summertime Canadian high Arctic lower troposphere"

_Atmospheric Chemistry and Physics, 2017_

## Referee Comment (RC1) · Anonymous Referee #1 · 8 Jun 2017

This work in particular presents aircraft-based measurement results of particle-phase TMA in summertime Arctic aerosols, and analyzed its origin in combing with other supporting data. Overall, the paper is well written and clearly presented, in particular, I found the results results and data interpretation were convincing, overall I recommend its publication in ACP, while a number of comments need to be addressed first, as listed below: (1) The paper in total analyzed 7412 particles. Is this number covering all the partciles sampled during the measurement? Or if you actually sampled more samples, then based on what principle, you chose this number of particles? Also, as you have sampled particles during 4-12 July, any changes of the properties with the changes of meterological conditions of each day? Any difference between daytime

and nighttime if flight time allows? (2) TMA-, Na/Cl-, EC-, and levoglucosan-containg particles in total occupied a bit less than 50% of all particles. However, the rest particles (>50%ïïjĽwere not discussed, how about their properties? (3) As analyses on 7412 particles in fact only covers a very small portion of ambient particles, and single particle analyses in principle is not a bulk analysis, I think such limitations should be mentioned clearly. On the other hand, why the results based on a small portion of particles are representative should be justified as well. (4) The analyses regarding the sources and processes of TMA-containing particles might be discussed in combination with data for other species. Is that possible? (5) Do the authors look at other amines besides TMA in the aerosols? Although TMA might be the most abundant low molecular weight amine, other amines can be present in the aerosols as well.

---

## Referee Comment (RC2) · Anonymous Referee #2 · 6 Jul 2017

The manuscript presents a very interesting and comprehensive study, reporting the chemical composition of Arctic summertime aerosol, focusing particularly on TMA presence, defining particle origin, size and distribution. The chemical data are combined with important meteorological parameters and elaborated by proper methodologies. The discussion is extensive, clear and effective. For these reason I suggest its publication in ACP. I have only some minor questions: Do the number of collected particles was similar at the different altitudes? (Maybe I missed this information) I think you should have a comparable number of particles collected at different altitude level in order to assure representativeness. Although the results are very interesting, you analyzed only 7412 particles. . .how this small number could be considered representative for summertime Arctic aerosol? You discussed TMA but there are other type of amines which were not investigated and which could help in source apportionment studies...why didn't you consider other compounds (eg. Amino acids)? Why did you analyze backward trajectories only on 4,7 and 8 July? It would be better to clarify these points in the respective sections.

---

## Referee Comment (RC3) · Anonymous Referee #3 · 10 Jul 2017

Köllner et al present aircraft-based single-particle mass spectrometry (SPMS) results from flights in July 2014 in the Canadian High Arctic during NETCARE. The main individual particle types observed included trimethylamine and levoglucosan-containing particles. This is an important dataset that provides the first insights into amines in the changing Arctic and greatly improves our knowledge of Arctic aerosol mixing state. Concerns and suggestions (including major concerns about particle type assignments and associated conclusions) are described below.

During revision, the authors should work to improve the writing, as there are some grammar mistakes throughout the manuscript, some of which are noted below. Also,

[Figure]

there are over 10 "paragraphs" that consist of 1-3 sentences each and do not represent full paragraphs with fully developed thoughts; these sentences should be incorporated into longer paragraphs in the revision. There are two in preparation manuscripts that are referenced in the manuscript – Bozem et al 2017 and Molleker et al 2017 – yet neither the full reference or manuscript are provided for review purposes.

Major Comments:

- In particular, I have concerns about the low fraction of particles that appear to have been classified/identified. While over 7000 particles were chemically analyzed in the study, the percentages discussed in Section 3.2 and shown in Figure 5 indicate that only 46% of the particles were classified/identified in the data analysis. This is very low and seems concerning in that other particle types may be missing in the analysis. To help better inform the reader, it would be useful in the methods section to state the number of particles chemically analyzed (mass spectra produced) by ALABAMA during the study, as well as the fraction with dual-polarity mass spectra and fraction classified by the CRISP software. In addition, the phrasing "Chemical classes" in Table 1 is confusing, as it appears in the text that these refer to marker species, rather than particle types; the authors might consider using the phrasing "ion markers" or similar to differentiate Tables 1 and 2, or the authors might consider combining Tables 1 and 2 for improved clarity.

- Page 1 Line 6, Figure 2, Section 2.5, and Page 19 Lines 4-5: These sentences, figure, and section detail the laboratory measurement of TMA. However, the mass spectrum of TMA has already been published, using 266 nm LDI single-particle mass spectrometry, by Angelino et al. (2001), and the other literature cited in Section 2.5 (Healy et al 2015, Rehbein et al 2011) simply cite Angelino et al. (2001) for this assignment, which isn't clear as written. Therefore, unfortunately, this does not represent a new result. The authors are encouraged to move Figure 2 and Section 2.5 to the supplemental and to remove the sentences highlighting this result in the abstract and conclusions.

- Table 1: It would be most useful to cite SPMS lab characterization studies when possible and include the citation (e.g. Angelino et al (2001)) in the table itself, as it appears there is room. Non-SPMS literature (e.g. #6, 10-12) is not appropriate to cite here for the assignment of ion markers. No SPMS literature is cited for EC, for example. When lab study literature is available, field study literature is less appropriate, as it often simply cites other lab studies (e.g., Healy et al 2015 and Rehbein et al 2011 simply cite Angelino et al 2001 for TMA ion markers, so all three references are not necessary; Corbin et al 2012 also cites Silva et al 1999 for the levoglucosan assignment), and including field + lab literature in a comprehensive manner has not been done here (nor would it be necessary). Silva et al (1999) is the only study cited that shows a SPMS mass spectrum of levoglucosan. In that work, m/z -73 does not appear to be discussed as a tracer, so it is not clear where this ion attribution is coming from (could m/z -73 correspond to glyoxylic acid?).

- For greater clarity for the non-SPMS reader, the authors are encouraged to label the "Na/Cl" particles as "sea spray aerosol", after defining and explaining the corresponding mass spectrum and citing SPMS laboratory studies of sea spray aerosol (e.g. Gard et al 1998 (Science), Prather et al 2013 (PNAS), Guasco et al 2014 (ES&T)). In the abstract and Section 3.2.2, the authors attribute peaks at m/z -45, -59, and -71/-73 to carbohydrates (stated as levoglucosan and cellulose), but no single-particle mass spectrometry sea spray aerosol literature is cited to support these assignments and attribution to organic carbon coating sea spray aerosol. Further Prather et al and Guasco et al (noted above here), show SPMS spectra of individual sea spray particles produced in the laboratory and internally mixed with organic carbon but do not mention these peaks. The authors are encouraged to view these papers to see if greater knowledge of the Arctic sea spray aerosol can be gained through comparison to these previously published laboratory studies of sea spray aerosol produced from biologically active waters. Also, the authors might look at Cahill et al 2015 (Analytical Chem.) and Pratt et al 2009 (Nat. Geosc.), which shows ATOFMS of primary biological particles and has some of these markers.
- I would argue that the lack of negative ion mass spectra in high RH environments, particularly marine environments (e.g. Spencer et al 2008 (JGR)), is common. Therefore, I think that the authors can more strongly state that the lack of negatives is likely due to particle water suppression of negative ion formation. Recently, Guasco et al 2014 (ES&T) observed m/z 19 (H3O+) in sea spray aerosol particles that did not produce negative ions, further supporting the previous work of Neubauer et al.

- Simoneit et al 1999 (Atmos. Environ.) showed that levoglucosan can be used as a tracer of biomass burning (BB), so it is confusing that the authors attribute the levoglucosan to pollution plumes (e.g., Page 11 Line 6) rather than BB. Hu et al 2013 (Sci. Rep.) published a manuscript, using levoglucosan as a tracer of BB, and showing a significant impact of BB in the Arctic. The authors should be aware, however, that levoglucosan is not a conservative tracer as it can degrade during transport (e.g. Hennigan et al 2010, GRL). Further, the authors should consider comparing the K-dominant spectra to SPMS BB mass spectra (e.g. Silva et al 1999 (ES&T), Hudson et al 2004 (JGR), Pratt et al 2011 (ACP)), as the K-dominant particles are likely BB in origin (although they can have significant organic aerosol coating still – see Pratt and Prather 2009, ES&T). Given the significant number of SPMS papers attributing K+ to BB, it is unclear why the authors state that "potassium in a single particle spectrum must be interpreted with caution" (Page 16, Line 11). It would be useful to conduct air mass trajectory analysis and look at satellite smoke/fire maps to investigate if transported BB plumes could be detected to support the observations discussed herein of the levoglucosan-containing and K-containing particles. I would encourage the authors to consider a broader particle class of BB particles.

- Ge et al 2011 (Atmos Environ) shows that TMA is also emitted from BB. Given that the non-K-containing TMA particles are primarily found within the BL (so a marine source makes sense) and the K- and levoglucosan-containing TMA particles are found at higher altitudes, consideration of TMA from BB may be warranted. Currently this potential source of TMA is not mentioned.

- m/z -26 (CN-) and -42 (CNO-) are typically attributed to organic nitrogen fragments (e.g. Fergenson et al 2004 (Analytical Chem.) and Pratt et al 2009 (Nat. Geosc.)), rather than cyanide (Table 2 and text), in SPMS. These are also common peaks in BB SPMS spectra (e.g. Silva et al 1999). On Page 11 Line 3, the authors cite Li et al 2000 for the assignment of cyanide, but that is not a SMPS paper.

- Page 11 Line 5: It is stated that there is no vertical dependence in the EC-containing particles, but do the authors have sufficient statistics to test this with only 138 EC particles detected over the entire study at all altitudes? How many at each altitude were detected?

- Figure 9 and associated text: Given the issue of mostly positive-only mass spectra, the authors are encouraged to consider analysis of only the dual-polarity mass spectra here, as MSA as a negative ion would be similarly affected compared to sulfate. Without knowledge of the fraction of total mass spectra not having negative ions, this analysis is difficult to interpret.

- For further support of the presence of SOA, the authors might consider looking at m/z -43 (Qin et al 2012, Atmos. Environ.). It would also be useful to show the non-K-containing TMA average mass spectrum, as the marine source of these particles is a big highlight of this work – and yet Figure 4 shows particles with a large K peak, which may be from BB, as discussed above.

Minor Comments:

- Page 1 Lines 6-9: This sentence can be clarified, as the comparisons are incomplete as written.

- Page 2 Lines 5-6: Fix sentence structure – particles are not a process.

- Page 2 Lines 9-10: Need references.

- Page 2 Line 9: Suggest changing "pollution sources within" to "pollution transport to", as the polar dome changes transport patterns, not the emissions themselves.

[Figure]

- Page 2 Line 10: Change "leading" to "leads".

- Page 2 Lines 12-15: Not all of this literature corresponds to nucleation papers as implied, and many recent Arctic aerosol growth papers are also missing here. Also citations are not properly included in the sentence (formatting issue).

- Page 2 Lines 29-31: Sentence is missing reference.

- Page 3 Line 23: Fix grammar – change "parameters, several trace gases as well" to "parameters, and several trace gases, as well".

- Figure 1 caption: Change "Image are" to "Image is".

- Figure 1: Flight paths are difficult to view on small inset.

- Page 4 Line 2: Delete sentence as it is unnecessary and not incorporated in a paragraph.

- Page 5 Line 5: Give size range of PSLs used.

- Page 5 Line 8: Suggest adding the phrase "of individual particles" after "bipolar mass spectra" for clarity for the non-SPMS reader.

- Page 7, Lines 6-8: This statement seems to contradict the statement on page 5 line 25.

- Page 8 Lines 6-7: Rather than "biogenic emissions", the authors might consider the phrase "Arctic oceanic and terrestrial emissions" here, as the data is not presented to support the current statement (and the results show near equal contributions from TMA- and levoglucosan-containing particles).

- Page 8 Line 21: Word "respectively" is misplaced.

- Figure 4 caption: Fix grammer – "spectrum" vs "spectra" – and repetition of this word.

- Table 2: For greater clarity, align the text in the right-most column with the ion assignments in the third column.

- Page 12 Lines 1-2: This sentence is redundant – essentially the repeat of the previous sentence.

- Page 12 Line 9: Sorensen et al 2005 is not appropriate here. The authors might also consider Gard et al 1998 (Science), which showed SPMS of sea spray aerosol aging.

- Page 12 Lines 11 & 13: "NaCl2" should be "NaCl2-" as these are ions.

- Figure 7 seems redundant and unnecessary considering the spectrum in Figure 4.

- Figure 8: This could be moved to the supplemental as it essentially repeats the information already in Figures 5 and 6, is more difficult to interpret, and has fewer particles included per bin to allow robust trends to be easily observed.

- Figure 10: To make the figure easier to read and interpret, the authors should consider showing as a fraction of TMA particles, rather than all particles, and also show the TMA particle numbers, rather than total particle numbers, as that information is already in Figure 5.

- Page 16, Line 2: Do the authors have sufficient statistics in all size bins to state that these particles are only from 280-380 nm?

- Page 17, Lines 4-5: If 8 July is a focus here, why did the authors choose not to show the wind data, even in the supplemental?

- Figure 14: Please define "potential emission sensitivity" in the caption or text. The units of seconds are not clear.

---

## Author Comment (AC1) · 12 Sep 2017

We thank Referee #1 for her/his comments and suggestions which helped to improve the manuscript. Our response is formatted as follows:

**Reviewer's comments**

Author's reply

Changes to the manuscript

All page, line, section and figure numbers in bold refer to the original manuscript, all others to the revised version.

**This work in particular presents aircraft-based measurement results of particle-phase TMA in summertime Arctic aerosols, and analyzed its origin in combing with other supporting data. Overall, the paper is well written and clearly presented, in particular, I found the results results and data interpretation were convincing, overall I recommend its publication in ACP, while a number of comments need to be addressed first, as listed below:**

**(1) The paper in total analyzed 7412 particles. Is this number covering all the partciles sampled during the measurement? Or if you actually sampled more samples, then based on what principle, you chose this number of particles?**

7412 is the total number of particles analyzed by ALABAMA during out-of-cloud sampling for measurements between 4 and 12 July. The inlet we used for particle sampling is not suitable for in-cloud measurements. Therefore, aerosol particles analyzed inside clouds (in total 763) were discarded by using measurements of an under wing FSSP (Forward Scattering Spectrometer) probe which measured cloud droplet number concentration (Leaitch et al., 2016).

**(2) Also, as you have sampled particles during 4-12 July, any changes of the properties with the changes of meterological conditions of each day?**

Prevailing high pressure influence in the Resolute Bay region leads to comparable meteorological conditions each day (during 4 – 12 July 2014).

**(3) Any difference between daytime and nighttime if flight time allows?**

We conducted no flights during nighttime. The earliest flight take-off time was at 15 UTC (local time: -5 h) and the latest flight landing time was at 23 UTC. Nevertheless, we would not expect large differences due to the small diurnal changes in angle of solar light during this time of the year.

**(4) TMA-, Na/Cl-, EC-, and levoglucosan-containg particles in total occupied a bit less than 50% of all particles. However, the rest particles (>50%) were not discussed, how about their properties?**

We particularly focused on the presence of particulate TMA in the summertime Arctic, including origin, size and vertical distribution. For this reason, we discussed only other particle types which either could confirm the marine origin of TMA (Na/Cl-cont. particles) or which could be clearly put into contrast to TMA in terms of size, vertical dependence and/or origin (e.g., levoglucosan-cont. particles).

However, a large fraction (29 %) of the particles prior classified as "others" contain potassium and sulfate. In the revised manuscript such particles are summarized as "K/S-containing". This particle type is introduced in Sect. 3.2. We discussed this particle type by analyzing the associated size (Fig. 5) and vertical distributions (Fig. 6) as well as the mean spectrum (Fig. 4e and Tab. 2) in Sect. 3.2.1 as follows (p.8, l.17 – p.11, l.7):

Levoglucosan, EC and potassium are known to be primarily produced from fossil fuel and biomass combustion processes (e.g., Bond et al., 2007; Simoneit, 2002; Andreae and Merlet, 2001; Simoneit et al., 1999). In particular, levoglucosan is formed via the breakdown of cellulose during biomass burning processes. The size distributions of levoglucosan- and EC-containing particles are shifted towards larger diameters compared to other particle types (Fig. 5). This result suggests these particles were exposed to chemical aging during long-range transport from biomass burning sources. K/S-containing particles are more evenly distributed across the size distribution (280 - 970 nm). Mean mass spectra of EC-, levoglucosan- and K/S-containing particles (Fig. 4c-e and Tab. 2) indicate a concurrent presence of sulfate (m/z -97/99 ($HSO_4^-$)), MSA (m/z -95 ($CH_3SO_3^-$)) and organic nitrogen compounds (m/z -26 ($CN^-$), m/z -42 ($CNO^-$)). Further given that the $K^+$ ion signals (m/z +39/41) are dominant in mean cation spectra (Fig. 4c-e), we can likely attribute these particles to a biomass burning source (e.g., Silva et al., 1999; Hudson et al., 2004; Pratt and Prather, 2009; Pratt et al., 2011). Furthermore, Zauscher et al. (2013) assigned negative ion signals at m/z -73 ($C_3H_5O_2^-$) to glyoxylic acid, which is typically present in biomass burning related SPMS spectra. Pratt et al. (2011) analyzed biomass burning particles internally mixed with oxalic acid (m/z -89 ($C_2O_4H^-$)). Both peaks are present in EC and levoglucosan mean mass spectra (Fig. 4c,d and Tab. 2). The vertical dependence in EC-containing particles is not further analyzed here due to the low statistical significance of 138 particles detected over the entire study at all altitudes. From the vertical profile of levoglucosan- and K/S-containing particles given in Fig. 6, it can be seen that their fractions increase with increasing altitude. These observations correspond to enhanced CO mixing ratios and $N_{d>250nm}$ (Fig. 3) providing further evidence for biomass burning as the source of levoglucosan- and K/S-containing particles. Despite the potential for oxidation of levoglucosan during transport, it has been previously reported as associated with biomass burning aerosol in Arctic regions (Hu et al., 2013; Fu et al., 2013, 2009).

The abstract and conclusive parts of the revised manuscript were modified accordingly.

Furthermore, the non-classified number of particles in "others" is reduced. We further added the mean spectrum of "others" in the Supplement (Fig. S7), which is mentioned in the main manuscript as follows (p.8, l.8 – 10):

The mean spectrum of the remaining 2039 particles (28 % of mass spectra analyzed by the ALABAMA), which could not be classified into one of the five particle groups outlined above, is shown in Fig. S7. For the further analysis we summarize these remaining particles in "others".

**(5) As analyses on 7412 particles in fact only covers a very small portion of ambient particles, and single particle analyses in principle is not a bulk analysis, I think such limitations should be mentioned clearly. On the other hand, why the results based on a small portion of particles are representative should be justified as well.**

We included discussions considering limitations by the number of analyzed particles and representativeness in the conclusive part of the revised manuscript as follows:

p.20, l.6 – p.21, l.3

SPMS measurements do not provide bulk analysis of aerosol chemical composition, therefore we can not obtain TMA mass concentrations. Nevertheless, the number of particles analyzed by the ALABAMA (> 7000) is sufficient to conduct a statistical analysis. This allows us to draw conclusions about mixing state, vertical and size distributions as well as potential emission sources of particulate TMA in summertime Arctic regions.

p. 22, l.17 - 20

This is the first study demonstrating the incorporation of amines in Arctic aerosol from inner-Arctic sources. Based on spatial and temporal limitations of our measurements, it is difficult to assess how representative our findings are of the broader Arctic region. However, recent measurements confirm the presence of particulate amines and its marine-biogenic source at another Arctic site (Alert, 82.5 °N) (Leaitch et al., 2017).

**(6) The analyses regarding the sources and processes of TMA-containing particles might be discussed in combination with data for other species. Is that possible?**

Thank you for this suggestion.

We discussed the ALABAMA data compared to concurrent data of the ToF-AMS (MSA mass concentrations as well as H/C and O/C ratio, see Willis et al., 2016, 2017) as well as CO mixing ratios. In particular, the comparison of particulate TMA with H/C and O/C ratios shows that TMA particles within the Arctic BL are less oxidative aged. This analysis provides further evidence for an inner-Arctic source of particulate TMA. Furthermore, the comparison with MSA mass concentration gives further indications for a marine-biogenic influence on "Non-K,NH$_4$-containing" TMA particles. The comparison with CO mixing ratios shows that TMA particles containing potassium and/or levoglucosan (likely from above the local BL) likely originate from biomass burning sources. The last point picks up a discussion suggested from Referee#3.

The new figures (Fig. 8,12,14) and associated discussions are included in the revised manuscript as follows:

p.13, l.10-15

Comparison of HR-ToF-AMS estimated oxygen-to-carbon (O/C) and hydrogen-to-carbon (H/C) ratios withthe ALABAMA particulate TMA fraction gives an indication of the degree of particle oxidative aging (e.g., Jimenez et al., 2009; Heald et al., 2010; Ng et al., 2011;Willis et al., 2017). Less oxygenated organics measured with the HR-ToF-AMS were present when the fraction of TMA-containing particles was high (Fig. 8a, up to 75 % in the upper left corner). This suggests that a large fraction of particulate TMA, especially within the BL (indicated with green circles in Fig. 8b), had not been subject to extensive oxidative aging.

p.16, l.2-4

Comparison between CO mixing ratios and TMA sub-types abundance (Fig. 12) shows larger fractions of "K,NH$_4$,S-containing" and "levoglucosan-containing" TMA particle sub-types in higher CO environments compared to "Non-K,NH$_4$-containing" TMA particles.

p.17, l.10-15

Furthermore, Fig. 14 indicates a positive correlation between MSA mass concentrations measured with HR-ToF-AMS and the fraction of "Non-K,NH$_4$-containing" TMA particles. Given that MSA can be used as an indicator for marine influence on sub-micron aerosol, we can conclude that the existence of an inner-Arctic marine-biogenic source of TMA is likely. Moreover, "Non-K,NH$_4$-containing" TMA particles are most abundant at the lowest altitudes (Fig. 11) and are coincident with the presence of less aged particulate organic aerosol (Fig. 8).

The abstract and conclusive parts of the revised manuscript were modified accordingly.

We further compared our measurements on 8 July with data presented in a companion paper from Burkart et al. (2017) as follows (Sect. 3.4, p.19, l. 33-35):

In addition, high organic-to-sulfate and MSA-to-sulfate ratios measured with the HR-ToF-AMS during this flight leg (see Sect. 4.3 in Burkart et al. (2017)) indicate that particle growth was driven by ocean-derived precursor gases (dimethylsulfide and organic species).

**(7) Do the authors look at other amines besides TMA in the aerosols? Although TMA might be the most abundant low molecular weight amine, other amines can be present in the aerosols as well.**

We investigated the presence of other alkylamines (other than TMA) and amino acid in ambient single particles. We found that none of this previously identified SPMS marker ions of other alkylamines and amino acid distinctively appear in our ambient Arctic mass spectra besides TMA.

This investigation and the associated discussion are added to the supplementary part of the revised manuscript (Sect. 4). We further added the following comment in the main manuscript (p.8, l.5-6):

Other alkylamines (other than TMA) and amino acids could not be identified (Supplement Sect. 4).

In addition to changes suggested by the referees, we did some minor changes in the revised manuscript as follows:

(1) The vertically resolved fraction of different particle types in figures 6 and 11 are now cumulative presented. This improves readability and makes the comparison between different particle types easier.
(2) Particles summarized as "others" appear now in figures 6 and 11.
(3) We unified axis notation in figures 5, 6, 10, 11, 12, 14 and 15.
(4) The colored flight tracks in Fig. 2 were partly wrongly assigned. We changed this.
(5) As described in the response to reviewer #1 comment #1, the inlet we used for aerosol sampling is not suitable for in-cloud measurements. Therefore, aerosol measurements inside clouds had been discarded. In the revised version of the manuscript, this selection had been made up for the vertical profiles (median and interquartile ranges) of $N_{d>5nm}$ and $N_{d>250nm}$ (Fig. 3).

**References:**

[revised manuscript text omitted]

---

## Author Comment (AC2) · 12 Sep 2017

We thank Referee #2 for her/his comments and suggestions which helped to improve the manuscript. Our response is formatted as follows:

**Reviewer's comments**

Author's reply

Changes to the manuscript

All page, line, section and figure numbers in bold refer to the original manuscript, all others to the revised version.

**The manuscript presents a very interesting and comprehensive study, reporting the chemical composition of Arctic summertime aerosol, focusing particularly on TMA presence, defining particle origin, size and distribution. The chemical data are combined with important meteorological parameters and elaborated by proper methodologies. The discussion is extensive, clear and effective. For these reason I suggest its publication in ACP. I have only some minor questions:**

(1) **Do the number of collected particles was similar at the different altitudes? (Maybe I missed this information) I think you should have a comparable number of particles collected at different altitude level in order to assure representativeness.**

The vertical profiles in **Fig. 7 and 8** show information of the total number of analyzed particles in each altitude on the left hand side. Relative fractions in **Fig. 11** refer to the total number already given in **Fig. 7**. However, **Fig. 11** had been changed anyway according to Referee#3. Now relative fractions in **Fig. 11** (Fig. 11) refer to the total number of TMA-containing particles in each altitude bin.

Numbers differ in different altitude levels between approximately 70 and 1200 **(Fig. 7),** 20 and 300 **(Fig. 8)** as well as 30 and 400 (Fig. 11). Nevertheless, the total particle number in each bin is sufficient to conduct statistical approaches.

(2) **Although the results are very interesting, you analyzed only 7412 particles. . .how this small number could be considered representative for summertime Arctic aerosol?**

We included discussions considering limitations by the number of analyzed particles and representativeness in the conclusive part of the revised manuscript as follows:

p.20, l.6 – p.21, l.3

SPMS measurements do not provide bulk analysis of aerosol chemical composition, therefore we can not obtain TMA mass concentrations. Nevertheless, the number of particles analyzed by the ALABAMA (> 7000) is sufficient to conduct a statistical analysis. This allows us to draw conclusions about mixing state, vertical and size distributions as well as potential emission sources of particulate TMA in summertime Arctic regions.

p. 22, l.17 – 20

This is the first study demonstrating the incorporation of amines in Arctic aerosol from inner-Arctic sources. Based on spatial and temporal limitations of our measurements, it is difficult to assess how representative our findings are of the broader Arctic region. However, recent measurements confirm the presence of particulate amines and its marine-biogenic source at another Arctic site (Alert, 82.5 °N) (Leaitch et al., 2017).

**(3) You discussed TMA but there are other type of amines which were not investigated and which could help in source apportionment studies. . .why didn't you consider other compounds (eg. Amino acids)?**

We investigated the presence of other alkylamines (other than TMA) and amino acid in ambient single particles. We found that none of this previously identified SPMS marker ions of other alkylamines and amino acid distinctively appear in our ambient Arctic mass spectra besides TMA.

This investigation and the associated discussion are added to the supplementary part of the revised manuscript (Sect. 4). We further added the following comment in the main manuscript (p.8, l.5-6):

Other alkylamines (other than TMA) and amino acids could not be identified (Supplement Sect. 4).

**(4) Why did you analyze backward trajectories only on 4,7 and 8 July? It would be better to clarify these points in the respective sections.**

This had been explained in the original manuscript (caption **Fig. 12/** Fig. 15) as follows:

Only time intervals with a total number of measured particles larger than 20 were considered. Measurements within the BL on 5, 10 and 12 July did not provide any 10-min time interval with more than 20 spectra.

In addition to changes suggested by the referees, we did some minor changes in the revised manuscript as follows:

**(1)** The vertically resolved fraction of different particle types in figures 6 and 11 are now cumulative presented. This improves readability and makes the comparison between different particle types easier.
**(2)** Particles summarized as "Others" appear now in figures 6 and 11.
**(3)** We unified axis notation in figures 5, 6, 10, 11, 12, 14 and 15.
**(4)** The colored flight tracks in Fig. 2 were partly wrongly assigned. We changed this.
**(5)** As described in the response to reviewer #1 comment #1, the inlet we used for aerosol sampling is not suitable for in-cloud measurements. Therefore, aerosol measurements inside clouds had been discarded. In the revised version of the manuscript, this selection had been made up for the vertical profiles (median and interquartile ranges) of $N_{d>5nm}$ and $N_{d>250nm}$ (Fig. 3).

**Reference:**

Leaitch, W. R., Russell, L. M., Liu, J., Kolonjari, F., Toom, D., Huang, L., Sharma, S., Chivulescu, A., Veber, D., and Zhang, W.: Organic Functional Groups in the Submicron Aerosol at 82.5∘ N from 2012 to 2014, Atmospheric Chemistry and Physics Discussions, 2017, 1–38, doi:10.5194/acp-2017-511, https://www.atmos-chem-phys-discuss.net/acp-2017-511/, 2017.

---

## Author Comment (AC3) · 12 Sep 2017

We thank Referee #3 for her/his comments and suggestions which helped to improve the manuscript. Our response is formatted as follows:

**Reviewer's comments**

Author's reply

Changes to the manuscript

All page, line, section and figure numbers in bold refer to the original manuscript, all others to the revised version.

**Köllner et al present aircraft-based single-particle mass spectrometry (SPMS) results from flights in July 2014 in the Canadian High Arctic during NETCARE. The main individual particle types observed included trimethylamine and levoglucosan-containing particles. This is an important dataset that provides the first insights into amines in the changing Arctic and greatly improves our knowledge of Arctic aerosol mixing state. Concerns and suggestions (including major concerns about particle type assignments and associated conclusions) are described below.**

(1) **During revision, the authors should work to improve the writing, as there are some grammar mistakes throughout the manuscript, some of which are noted below. Also, there are over 10 "paragraphs" that consist of 1-3 sentences each and do not represent full paragraphs with fully developed thoughts; these sentences should be incorporated into longer paragraphs in the revision. There are two in preparation manuscripts that are referenced in the manuscript – Bozem et al 2017 and Molleker et al 2017 – yet neither the full reference or manuscript are provided for review purposes.**

We corrected grammar mistakes which Referee #3 outlined below. Furthermore, the recent manuscript had been proofread with regard on spelling and grammar by English native speakers among the co-authors. We further incorporated several sentences into longer paragraphs. Together, such changes are highlighted in red (color refers to "own revisions") in the "tracked-changes" version.

We removed both references which are in preparation. If these manuscripts will be in review process before publication of our manuscript, we will add them.

**Major comments:**

(2) **In particular, I have concerns about the low fraction of particles that appear to have been classified/identified. While over 7000 particles were chemically analyzed in the study, the percentages discussed in Section 3.2 and shown in Figure 5 indicate that only 46% of the particles were classified/identified in the data analysis. This is very low and seems concerning in that other particle types may be missing in the analysis.**

We particularly focused on the presence of particulate TMA in the summertime Arctic, including origin, size and vertical distribution. For this reason, we discussed only other particle types which either could confirm the marine origin of TMA (Na/Cl-cont.

particles) or which could be clearly put into contrast to TMA in terms of size, vertical dependence and/or origin (e.g., levoglucosan-cont. particles).

However, a large fraction (29 %) of the particles prior classified as "others" contain potassium and sulfate. In the revised manuscript such particles are summarized as "K/S-containing". This particle type is introduced in Sect. 3.2. We discussed this particle type by analyzing the associated size and vertical distributions as well as the mean spectrum (Fig. 4e and Tab. 2) in Sect. 3.2.1 as follows (p.8, l.17 – p.11, l.7):

Levoglucosan, EC and potassium are known to be primarily produced from fossil fuel and biomass combustion processes (e.g., Bond et al., 2007; Simoneit, 2002; Andreae and Merlet, 2001; Simoneit et al., 1999). In particular, levoglucosan is formed via the breakdown of cellulose during biomass burning processes. The size distributions of levoglucosan- and EC-containing particles are shifted towards larger diameters compared to other particle types (Fig. 5). This result suggests these particles were exposed to chemical aging during long-range transport from biomass burning sources. K/S-containing particles are more evenly distributed across the size distribution (280 - 970 nm). Mean mass spectra of EC-, levoglucosan- and K/S-containing particles (Fig. 4c-e and Tab. 2) indicate a concurrent presence of sulfate (m/z -97/99 ($HSO_4^-$)), MSA (m/z -95 ($CH_3SO_3^-$)) and organic nitrogen compounds (m/z -26 ($CN^-$), m/z -42 ($CNO^-$)). Further given that the $K^+$ ion signals (m/z +39/41) are dominant in mean cation spectra (Fig. 4c-e), we can likely attribute these particles to a biomass burning source (e.g., Silva et al., 1999; Hudson et al., 2004; Pratt and Prather, 2009; Pratt et al., 2011). Furthermore, Zauscher et al. (2013) assigned negative ion signals at m/z -73 ($C_3H_5O_2^-$) to glyoxylic acid, which is typically present in biomass burning related SPMS spectra. Pratt et al. (2011) analyzed biomass burning particles internally mixed with oxalic acid (m/z -89 ($C_2O_4H^-$)). Both peaks are present in EC and levoglucosan mean mass spectra (Fig. 4c,d and Tab. 2). The vertical dependence in EC-containing particles is not further analyzed here due to the low statistical significance of 138 particles detected over the entire study at all altitudes. From the vertical profile of levoglucosan- and K/S-containing particles given in Fig. 6, it can be seen that their fractions increase with increasing altitude. These observations correspond to enhanced CO mixing ratios and $N_{d>250nm}$ (Fig. 3) providing further evidence for biomass burning as the source of levoglucosan- and K/S-containing particles. Despite the potential for oxidation of levoglucosan during transport, it has been previously reported as associated with biomass burning aerosol in Arctic regions (Hu et al., 2013; Fu et al., 2013, 2009).

The abstract and conclusive parts of the revised manuscript were modified accordingly.

Furthermore, the non-classified number of particles in "others" is reduced. We further added the mean spectrum of "others" in the Supplement (Fig. S7), which is mentioned in the main manuscript as follows (p.8, l.8 – 10):

The mean spectrum of the remaining 2039 particles (28 % of mass spectra analyzed by the ALABAMA), which could not be classified into one of the five particle groups outlined above, is shown in Fig. S7. For the further analysis we summarize these remaining particles in "others".

(3) **To help better inform the reader, it would be useful in the methods section to state the number of particles chemically analyzed (mass spectra produced) by ALABAMA during the study, as well as the fraction with dual-polarity mass spectra and fraction classified by the CRISP software.**

We replaced information about the number of particles chemically analyzed by ALABAMA from Sect. 3.2 to Sect. 2.4 (Methods) and added the fraction with dual-polarity mass spectra (p.5, l.26-27) as follows:

In total, 7412 particles were chemically analyzed (mass spectra produced) by the ALABAMA during the study. 94 % of these mass spectra include size information. 80 % of these mass spectra have dual-polarity.

The fraction of mass spectra classified/identified into five particle types is added (p.8, l. 2-3, Sect. 3.2.) as follows:

Applying the marker method (Sect. 2.4), we classified 5373 particle mass spectra (72 % of the mass spectra analyzed by ALABAMA (Sect. 2.4)) into five distinct particle types: TMA-, Na/Cl-, EC-, levoglucosan- and K/S-containing particles.

**(4) In addition, the phrasing "Chemical classes" in Table 1 is confusing, as it appears in the text that these refer to marker species, rather than particle types; the authors might consider using the phrasing "ion markers" or similar to differentiate Tables 1 and 2, or the authors might consider combining Tables 1 and 2 for improved clarity**

We re-formulated "Chemical class" by "Marker species" and "Marker conditions" by "Ion markers" in Tab. 1 and in several parts in the text. Thus, the caption of Tab. 1 had been changed.

**(5) Page 1 Line 6, Figure 2, Section 2.5, and Page 19 Lines 4-5: These sentences, figure, and section detail the laboratory measurement of TMA. However, the mass spectrum of TMA has already been published, using 266 nm LDI single-particle mass spectrometry, by Angelino et al. (2001), and the other literature cited in Section 2.5 (Healy et al 2015, Rehbein et al 2011) simply cite Angelino et al. (2001) for this assignment, which isn't clear as written. Therefore, unfortunately, this does not represent a new result. The authors are encouraged to move Figure 2 and Section 2.5 to the supplemental and to remove the sentences highlighting this result in the abstract and conclusions.**

**Section 2.5** and **Fig. 2** had been moved to the supplementary part (Sect. 2) and corresponding sentences in the abstract and conclusions had been deleted.

We further changed the part of citations as follows (Supplement Sect. 2):

These measurements confirm previous laboratory observations using 266 nm laser desorption/ionization single-particle mass spectrometry (Angelino et al., 2001). Based on the study of Angelino et al. (2001), particulate TMA have been previously detected in ambient air using SPMS (e.g. Roth et al., 2016; Healy et al., 2015; Rehbein et al., 2011).

We further added a sentence to this topic in the main manuscript (p.6, l.6-7):

The identification of ion markers m/z +59 and +58 for TMA by Angelino et al. (2001) was confirmed by additional laboratory measurements with the ALABAMA (Supplement Sect. 2).

**(6) Table 1: It would be most useful to cite SPMS lab characterization studies when possible and include the citation (e.g. Angelino et al (2001)) in the table itself, as it appears there is room. Non-SPMS literature (e.g. #6, 10-12) is not appropriate to cite here for the assignment of ion markers. No SPMS literature is cited for EC, for example. When lab study literature is available, field study literature is less appropriate, as it often simply cites other lab studies (e.g., Healy et al 2015 and Rehbein et al 2011 simply cite Angelino et al 2001 for TMA ion markers, so all three references are not necessary; Corbin et al 2012 also cites Silva et al 1999 for the levoglucosan assignment), and including field + lab literature in a comprehensive manner has not been done here (nor would it be necessary).**

We used non-SPMS literature for the assignment of certain substances to particles sources (e.g., levoglucosan to biomass burning) rather than assignment of SPMS ion markers to substances. However, to make this point more clear, we put the non-SPMS literature directly behind the related subjects. We further replaced the row including listed references for the assignment of ion markers to substances next to the list of ion markers.

Furthermore, we agree that lab studies are mandatory for ion marker assignment. Nevertheless, we think that conducted field studies with particle assignment based on those lab studies are necessary to mention in order to highlight the widely-spread acceptance in the SPMS community and that also other SPMS were able to detect those peaks. In order to make a clearer differentiation between lab and field studies, we separated those in the reference row. Subsequently, the caption of Tab. 1 had been changed as follows:

Marker species (with acronyms) and associated ion markers used in this study. Further given are references (SPMS lab and field studies) used for the assignment of ion markers as well as additional comments on marker species and ions.

We keep the citations as enumerated footnotes below the table. Otherwise, the table would be to full and would contain many repetitions. Furthermore, Schmidt et al. (2017) conducted lab studies with ALABAMA including measurements of elemental carbon.

**(7) Silva et al (1999) is the only study cited that shows a SPMS mass spectrum of levoglucosan. In that work, m/z -73 does not appear to be discussed as a tracer, so it is not clear where this ion attribution is coming from (could m/z -73 correspond to glyoxylic acid?).**

There is no consensus in the SPMS literature about the marker ions for levoglucosan. Laboratory mass spectra (Silva et al. (1999) and our own data) show -45, -59 and -71, while others (Moffet et al. (2008) and Corbin et al. (2012)) also used -73 as a marker ion. We conducted the levoglucosan analysis with and without -73 and found that the difference is only 2 %. Thus, the main fraction of levoglucosan-containing particle spectra includes m/z -73. According to Zauscher et al. (2013), ion signals at m/z -73 can be assigned to glyoxylic acid, which likely originate from biomass burning. We decided not using m/z -73 as ion marker for levoglucosan; therefore fractions of levoglucosancontaining particle types and tables 1 and 2 changed slightly. The discussion including ion signals at m/z -73 can be found in Sect. 3.2.1 (p.10, l.1 – p.11, l.2):

Furthermore, Zauscher et al. (2013) assigned negative ion signals at m/z -73 ($C_3H_5O_2^-$) to glyoxylic acid, which is typically present in biomass burning related SPMS spectra. Pratt et al. (2011) analyzed biomass burning particles internally mixed with oxalic acid (m/z -89 ($C_2O_4H^-$)). Both peaks are present in EC and levoglucosan mean mass spectra (Fig. 4c,d and Tab. 2).

**(8) For greater clarity for the non-SPMS reader, the authors are encouraged to label the "Na/Cl" particles as "sea spray aerosol", after defining and explaining the corresponding mass spectrum and citing SPMS laboratory studies of sea spray aerosol (e.g. Gard et al 1998 (Science), Prather et al 2013 (PNAS), Guasco et al 2014 (ES&T)).**

Our intention here was to name the particle types by the ion fragments used for the marker method rather than name them by their potential sources. The potential source of Na/Cl-containing particles is discussed in Sect 3.2.2. We want to be consistent in the manuscript and therefore keep the particle type notation also after discussing the source.

Literature, noted above, had been added to the manuscript.

**(9) In the abstract and Section 3.2.2, the authors attribute peaks at m/z -45, -59, and -71/-73 to carbohydrates (stated as levoglucosan and cellulose), but no single-particle mass spectrometry sea spray aerosol literature is cited to support these assignments and attribution to organic carbon coating sea spray aerosol. Further Prather et al and Guasco et al (noted above here), show SPMS spectra of individual sea spray particles produced in the laboratory and internally mixed with organic carbon but do not mention these peaks. The authors are encouraged to view these papers to see if greater knowledge of the Arctic sea spray aerosol can be gained through comparison to these previously published laboratory studies of sea spray aerosol produced from biologically active waters. Also, the authors might look at Cahill et al 2015 (Analytical Chem.) and Pratt et al 2009 (Nat. Geosc.), which shows ATOFMS of primary biological particles and has some of these markers.**

Thank you for catching up this point and suggesting different literature to view.
We added (amongst others) Prather et al. (2013), Guasco et al. (2014), Pratt et al. (2009a) in the revised discussion.

We further re-formulated this part as follows (Sect. 3.2.2, p.12, l.12 – p. 13, l.5):

Interestingly, some of the Na/Cl-containing particles are internally mixed with different inorganics (such as magnesium and calcium) as well as oxygen- and nitrogen-containing organic compounds, as indicated by the mean spectrum in Fig. 4b and Fig. 7. It is known from previous SPMS laboratory studies on sea spray particles produced from biologically active waters that magnesium, calcium and organic nitrogen species present on inorganic salts can be arise from biological activity (Prather et al., 2013; Guasco et al., 2014). In particular, organic nitrogen fragments together with calcium, sodium and phosphate have been linked to signatures of biological species (e.g., Pratt et al., 2009a; Schmidt et al.,

2017). SPMS spectra of biological particles in Pratt et al. (2009a) further indicate the occurrence of oxygen-containing organic compounds at m/z -71 ($C_3H_3O_2^-$). Laboratory studies with the ALABAMA investigating biological species (such as bacteria and pollen) also showed the existence of negative ion signals at m/z -45 ($C_2H_5O^-$), m/z -59 ($C_3H_7O_2^-/C_3H_9N^-$) and m/z -71 ($C_3H_3O_2^-/C_4H_7O^-$) in addition to the presence of phosphate and organic nitrogen compounds (Schmidt et al., 2017). Anion signals at m/z -26 ($C_2H_2^-$) and m/z -42 ($C_2H_2O^-/C_3H_6^-$) can be further attributed to cellulose (Schmidt et al., 2017). Moreover, Trimborn et al. (2002) reported the concurrent presence of sodium, chloride and oxygen-containing organic compounds (m/z -73 ($C_3H_5O_2^-$) and m/z -59 ($C_2H_3O_2^-$)) in ambient SPMS spectra and attributed them to organic containing sea salt particles. Other Non-SPMS studies (e.g., X-ray microscopy methods) have reported the occurrence of organic-rich (e.g., carboxylate) sea spray particles originating from microorganisms and organic compounds enriched in the sea surface microlayer in mid-latitude oceans (e.g., Quinn et al., 2014; Blanchard and Woodcock, 1980) and in Arctic regions (e.g., Wilson et al., 2015; Frossard et al., 2014; Hawkins and Russell, 2010; Russell et al., 2010). Taken together, the presence of magnesium and calcium together with nitrogen- and oxygen-containing organic species in sea spray particles suggests that such organic fragments have a marine-biogenic origin.

The associated parts in the abstract and conclusions had been removed.

**(10)** **I would argue that the lack of negative ion mass spectra in high RH environments, particularly marine environments (e.g. Spencer et al 2008 (JGR)), is common. Therefore, I think that the authors can more strongly state that the lack of negatives is likely due to particle water suppression of negative ion formation. Recently, Guasco et al 2014 (ES&T) observed m/z 19 (H3O+) in sea spray aerosol particles that did not produce negative ions, further supporting the previous work of Neubauer et al.**

We added the suggested literature in the supplementary discussion. We further added a brief comment on the lack of negative ions in the revised main manuscript (Sect. 2.4) as follows (p.5, l.27-30):

Considering the 20 % single-polarity spectra, potential reasons for the lack of negative ions are discussed in the Supplement Sect. 1. Briefly, it is likely that single-polarity spectra are produced in high relative humidity (RH) environments (Neubauer et al., 1998; Spencer et al.,2008), in particular marine environments (Guasco et al., 2014).

**(11)** **Simoneit et al 1999 (Atmos. Environ.) showed that levoglucosan can be used as a tracer of biomass burning (BB), so it is confusing that the authors attribute the levoglucosan to pollution plumes (e.g., Page 11 Line 6) rather than BB.**

This sentence had been anyway re-formulated with regard on your comment #42 (see reply to comment #42). However, biomass burning influenced air mass can be considered as polluted air within the clean summertime Arctic.

**(12)** **Hu et al 2013 (Sci. Rep.) published a manuscript, using levoglucosan as a tracer of BB, and showing a significant impact of BB in the Arctic. The authors should be aware,**

**however, that levoglucosan is not a conservative tracer as it can degrade during transport (e.g. Hennigan et al 2010, GRL).**

Thank you for catching this important aspect. According to Hennigan et al. (2010), the degradation of levoglucosan depends upon OH concentration. If levoglucosan is exposed to $1x10^6$ molec/ccm (OH global annual mean (Spivakovsky et al.,2000)) it has a mean lifetime of 1.1 days (range: 0.7-2.2 days) (Hennigan et al., 2010). We expect a longer lifetime for levoglucosan in sub-Arctic and high Arctic regions due to the lower OH concentration in these regions (June/July >65°N between 0.1 and $1x10^6$ molec/ccm (Spivakovsky et al.,2000 and Bahm and Khalil, 2004)).

However, we added the following sentence in the revised manuscript (Sect. 3.2.1, p. 11, l.7-8):

Despite the potential for oxidation of levoglucosan during transport, it has been previously reported as associated with biomass burning aerosol in Arctic regions (Hu et al., 2013; Fu et al., 2013, 2009).

**(13)    Further, the authors should consider comparing the Kdominant spectra to SPMS BB mass spectra (e.g. Silva et al 1999 (ES&T), Hudson et al 2004 (JGR), Pratt et al 2011 (ACP)), as the K-dominant particles are likely BB in origin (although they can have significant organic aerosol coating still – see Pratt and Prather 2009, ES&T).**

We agree that dominant potassium ion signals most likely originate from biomass burning. This information had been added in Tab. 1 as follows:

K-dominant SPMS spectra associated (K) with BB particles

However, we identified particle types on basis of their signal intensities of certain compounds above an ion peak area threshold. This method makes it difficult to state whether a peak is dominant.

Nevertheless, mean spectra of levoglucosan-, EC- and K/S-containing particles present in all cases a dominant potassium signal. A further analysis showed that all levoglucosan particles contain potassium. We added this discussion in Sect. 3.2.1 (p.9, l.3 – p.14, l.1):

Further given that the $K^+$ ion signals (m/z +39/41) are dominant in mean cation spectra (Fig. 4c-e), we can likely attribute these particles to biomass burning soruce (e.g., Silva et al., 1999; Hudson et al., 2004; Pratt and Prather, 2009; Pratt et al., 2011).

Furthermore, the revised manuscript includes a broader discussion on TMA-containing particles additionally composed of potassium (see comment #17).

The abstract and conclusive parts in the revised manuscript were modified accordingly.

**(14)    Given the significant number of SPMS papers attributing K+ to BB, it is unclear why the authors state that "potassium in a single particle spectrum must be interpreted with caution" (Page 16, Line 11).**

We deleted this sentence.

**(15)** **It would be useful to conduct air mass trajectory analysis and look at satellite smoke/fire maps to investigate if transported BB plumes could be detected to support the observations discussed herein of the levoglucosan-containing and K-containing particles.**

The focus of the paper is not long-range transport of biomass burning influenced air masses. These other particle types are added for reason of completeness and to show differences in size and vertical dependence of TMA-containing particles.

**(16)** **I would encourage the authors to consider a broader particle class of BB particles.**

We identified another particle type "K/S-containing" out of the prior "others" group (see comment #2).

We did not consider a more detailed classification of levoglucosan-containing particles with regard on potassium since: firstly, every spectrum contains potassium (see comment #13) and secondly, we think this approach is out of scope for this paper.

**(17)** **Ge et al 2011 (Atmos Environ) shows that TMA is also emitted from BB. Given that the non-K-containing TMA particles are primarily found within the BL (so a marine source makes sense) and the K- and levoglucosan-containing TMA particles are found at higher altitudes, consideration of TMA from BB may be warranted. Currently this potential source of TMA is not mentioned**

Thank you for pointing out this aspect. We discussed this additional source and explanation in the revised manuscript (Sect. 3.3) as follows (p.14, l.15 - p.16, l.6):

As can be seen in Fig. 9, a large fraction of TMA-containing particles (74 %) are additionally composed of biomass burning tracers such as potassium (67 %) and levoglucosan (7 %). According to Pöhlker et al. (2012), this internal mixture can be explained by potassium-containing particles acting as seeds for the condensation of organic material. Thus, the measured particulate TMA can be considered a secondary component that condensed on pre-existing primary particles. It is also conceivable that TMA particles containing potassium and levoglucosan are a result of biomass burning emissions (Schade and Crutzen, 1995; Ge et al., 2011a; Silva et al., 1999; Hudson et al., 2004; Pratt and Prather, 2009; Pratt et al., 2011). The size distribution of the TMA particles containing levoglucosan is shifted towards larger diameters compared to other TMA particle sub-types (Fig. 10). Moreover, Fig. 11 demonstrates that TMA particle sub-types including potassium and levoglucosan were more abundant above the BL in contrast to "Non-K,NH$_4$-containing" TMA particles. Comparison between CO mixing ratios and TMA sub-types abundance (Fig. 12) shows larger fractions of "K,NH$_4$,S-containing" and "levoglucosan-containing" TMA particle sub-types in higher CO environments compared to "Non-K,NH4-containing" TMA particles. Taken together, these results suggest that TMA particles containing levoglucosan and potassium likely originated from remote biomass burning emission sources and were transported to our measurement site.

The conclusions section and the abstract were modified accordingly.

**(18)    m/z -26 (CN-) and -42 (CNO-) are typically attributed to organic nitrogen fragments (e.g. Fergenson et al 2004 (Analytical Chem.) and Pratt et al 2009 (Nat. Geosc.)), rather than cyanide (Table 2 and text), in SPMS.**

We replaced the attribution of m/z -26 and -42 to "cyanide" by "nitrogen-containing organics" in Tab. 2 and in the text in sections 3.2.1 and 3.2.2. We further added the following references for the assignment of m/z -26 and -42 to "nitrogen-containing organics" (Tab. 2):  Pratt et al. (2009a), Silva et al. (1999), Fergenson et al. (2004), Prather et al. (2013) and Guasco et al. (2014).

**(19)    These are also common peaks in BB SPMS spectra (e.g. Silva et al 1999). On Page 11 Line 3, the authors cite Li et al 2000 for the assignment of cyanide, but that is not a SMPS paper.**

We replaced the citation Li et al. (2000) by Silva et al. (1999).

**(20)    Page 11 Line 5: It is stated that there is no vertical dependence in the EC-containing particles, but do the authors have sufficient statistics to test this with only 138 EC particles detected over the entire study at all altitudes? How many at each altitude were detected?**

We agree that the low number of EC-containing particles detected in different altitudes makes it difficult to interpret the data with regard on vertical dependence. We added the following comment in the revised manuscript (p.111, l.3-4):

The vertical dependence in EC-containing particles is not further analyzed here due to the low statistical significance of 138 particles detected over the entire study at all altitudes.

**(21)    Figure 9 and associated text: Given the issue of mostly positive-only mass spectra, the authors are encouraged to consider analysis of only the dual-polarity mass spectra here, as MSA as a negative ion would be similarly affected compared to sulfate. Without knowledge of the fraction of total mass spectra not having negative ions, this analysis is difficult to interpret.**

We agree that a prior differentiation between spectra with dual- and single-polarity is necessary to discuss the presence of MSA (similar to sulfate). Thus, we added a query for the presence of dual-polarity mass spectra in the classification scheme of TMA internal mixing state (Fig. 9). Subsequently, fractions and notation of particle sub-types as well as the description of the classification scheme in Sect 3.3 changed. Furthermore, figures 10, 11 and 15 had to be adjusted.

**(22)    For further support of the presence of SOA, the authors might consider looking at m/z -43 (Qin et al 2012, Atmos. Environ.).**

We found that approximately 51 % of TMA particle spectra include ion signals at m/z +43 and mz +27. Schmidt et al. (2017) reported that these peaks result from SOA formation from α-pinene ozonolysis in the laboratory. However, we are not sure to which extent α-pinene ozonolysis in the laboratory is comparable with Arctic marine SOA formation.

**(23)    It would also be useful to show the non-Kcontaining TMA average mass spectrum, as the marine source of these particles is a big highlight of this work – and yet Figure 4 shows particles with a large K peak, which may be from BB, as discussed above.**

Thank you for this suggestion. We added a new figure (Fig. 13) presenting the mean spectrum of the "Non-K,NH$_4$-containing" TMA particles. Due to the new prior differentiation between dual- and single-polarity spectra, the TMA particle sub-type "Non-K,NH$_4$-containing" includes just single-polarity spectra (12 %). The missing 6 % are separately listed in dual-polarity. Since the latter group is below 7 %, we did not consider this group for the further analysis. This requires more explanation which had been done as follows (Sect. 3.3., p.16, l.8 – p.17, l.10):

This is consistent with results from particle size distributions of TMA sub-types in Fig. 10 illustrating that the fractional abundance of "Non-K,NH$_4$-containing" TMA particles is highest between 280 and 380 nm compared to other sub-types containing levoglucosan and/or potassium. In particular, positive ion mass spectra of the sub-type "Non-K,NH$_4$-containing" (12 % single-polarity (yellow box in Fig. 9) and 6 % dual-polarity (not colored in Fig. 9)) show ion signals only for carbon cluster ions and fragments of hydrocarbons (Fig. 13a,b). Due to a suppression of anion signals, likely in high RH environments (Supplement Sect. 1), we cannot state whether sulfate or MSA were present in these particles. However, the dual-polarity mean spectrum of the 6 % TMA-containing particles not including potassium and ammonium (Fig. 13b, not colored in Fig. 9) indicates the concurrent presence of sulfate or MSA. From the absence of ammonium in these TMA particles containing sulfate or MSA, we can further conclude that aminium salts were present. This result demonstrates that amines, in addition to ammonia, may take part in the neutralization of acidic aerosol. This is of particular interest considering the reduced sources of ammonia in the Arctic and the ocean as a net sink of NH3 in the summertime Canadian Arctic (Wentworth et al., 2016).

**Minor Comments:**

**(24)    Page 1 Lines 6-9: This sentence can be clarified, as the comparisons are incomplete as written.**

We changed this sentence as follows:

Second, compared to particles observed aloft, TMA particles were smaller and less oxidized.

**(25)    Page 2 Lines 5-6: Fix sentence structure – particles are not a process.**

We changed the sentence structure as follows:

Among the processes driving Arctic warming, direct and indirect radiative effects of aerosol particles play a key role.

**(26)     Page 2 Lines 9-10: Need references.**

The listed literature at the end of the paragraph in the original manuscript refers to the last 3-4 sentences. However, this had been changed in the revised manuscript.

**(27)     Page 2 Line 9: Suggest changing "pollution sources within" to "pollution transport to", as the polar dome changes transport patterns, not the emissions themselves.**

This sentence had been changed as follows:

First, pollution sources within the polar dome are reduced during summer, since the polar dome surface extent is smaller during summer compared to winter.

**(28)     Page 2 Line 10: Change "leading" to "leads".**

This has been changed to "lead".

**(29)     Page 2 Lines 12-15: Not all of this literature corresponds to nucleation papers as implied, and many recent Arctic aerosol growth papers are also missing here. Also citations are not properly included in the sentence (formatting issue).**

Thanks for pointing out the formatting issue. This has been corrected.

We agree that not all of the literature correspond to nucleation papers. The listed literature at the end of the paragraph refers to the last 3-4 sentences (see comment #26). This has been changed anyhow.

We added in the row several recent studies focusing on new particle formation and aerosol growth in Arctic region: Leaitch et al. (2013), Wentworth et al. (2015) and Croft et al. (2016b).

**(30)     Page 2 Lines 29-31: Sentence is missing reference.**

The corresponding references can be found after the following sentence: Murphy et al., 2007.

**(31)    Page 3 Line 23: Fix grammar – change "parameters, several trace gases as well" to "parameters, and several trace gases, as well"**

We changed this phrase as suggested.

**(32)    Figure 1 caption: Change "Image are" to "Image is"**

This has been changed.

**(33)    Figure 1: Flight paths are difficult to view on small inset.**

The revised manuscript includes a separate figure showing flight tracks (Fig. 2).

**(34)    Page 4 Line 2: Delete sentence as it is unnecessary and not incorporated in a paragraph.**

This sentence has been removed.

**(35)    Page 5 Line 5: Give size range of PSLs used.**

We added this information as follows:

By comparing these values with the velocity of manufactured monodisperse polystyrene latex particles in five sizes ranging from 190 to 800 nm, we can derive the particle vacuum aerodynamic diameter ($d_{va}$).

**(36)    Page 5 Line 8: Suggest adding the phrase "of individual particles" after "bipolar mass spectra" for clarity for the non-SPMS reader.**

We added this phrase as suggested.

**(37)    Page 7, Lines 6-8: This statement seems to contradict the statement on page 5 line 25.**

We re-formulated this sentence as follows (Supplement Sect. 2):

Despite the higher laser ablation energy deployed during NETCARE 2014 compared to lab measurements, the molecular ion of TMA (m/z +59 (($CH_3)_3N$)) in combination with ions at m/z +58 ($C_3H_8N$) were produced during laser desorption/ionization (LDI) process.

**(38)    Page 8 Lines 6-7: Rather than "biogenic emissions", the authors might consider the phrase "Arctic oceanic and terrestrial emissions" here, as the data is not presented to support the current statement (and the results show near equal contributions from TMA- and levoglucosan-containing particles).**

This had been re-formulated as suggested.

**(39)    Page 8 Line 21: Word "respectively" is misplaced.**

The word had been re-placed to the end of sentence.

**(40)    Figure 4 caption: Fix grammer – "spectrum" vs "spectra" – and repetition of this word.**

We changed the caption of Fig. 4 as follows:

Bipolar mean spectra of the identified particle types: (a) TMA-containing (1688 particles = 23 %), (b) Na/Cl-containing (106 particles = 1 %), (c) EC-containing (138 particles = 2 %), (d) levoglucosan-containing (1312 particles = 18 %) and (e) K/S-containing (2129 particles = 29 %).

**(41)    Table 2: For greater clarity, align the text in the right-most column with the ion assignments in the third column.**

This improves readability and has been changed as suggested.

**(42)    Page 12 Lines 1-2: This sentence is redundant – essentially the repeat of the previous sentence.**

We changed this part as follows (p.11, l.3-8):

The vertical dependence in EC-containing particles is not further analyzed here due to the low statistical significance of 138 particles detected over the entire study at all altitudes. From the vertical profile of levoglucosan- and K/S-containing particles given in Fig. 6, it can be seen that their fractions increase with increasing altitude. These observations correspond to enhanced CO mixing ratios and $N_{d>250nm}$ (Fig. 3) providing further evidence for biomass burning as the source of levoglucosan- and K/S-containing particles. Despite the potential for oxidation of levoglucosan during transport, it has been previously reported as associated with biomass burning aerosol in Arctic regions (Hu et al., 2013; Fu et al., 2013, 2009).

**(43)** **Page 12 Line 9: Sorensen et al 2005 is not appropriate here. The authors might also consider Gard et al 1998 (Science), which showed SPMS of sea spray aerosol aging.**

We added this reference.

**(44)** **Page 12 Lines 11 & 13: "NaCl2" should be "NaCl2-" as these are ions**

Thank you for catching this error. We changed this in the revised manuscript.

**(45)** **Figure 7 seems redundant and unnecessary considering the spectrum in Figure 4.**

The large ion signal of sodium in the positive mean spectrum makes it difficult to identify smaller peaks in the negative mean spectrum (Fig. 4b). Thus, we think it is necessary to present Fig. 7 to show small negative ion signals.

**(46)** **Figure 8: This could be moved to the supplemental as it essentially repeats the information already in Figures 5 and 6, is more difficult to interpret, and has fewer particles included per bin to allow robust trends to be easily observed.**

This figure had been moved to the Supplement (Fig. S8).

**(47)** **Figure 10: To make the figure easier to read and interpret, the authors should consider showing as a fraction of TMA particles, rather than all particles, and also show the TMA particle numbers, rather than total particle numbers, as that information is already in Figure 5.**

We agree that the readability of figures 10 and 11 improves by presenting data as fraction of TMA-containing particles and including the total number of TMA-containing particles. Therefore, we changed both figures as suggested.

**(48)** **Page 16, Line 2: Do the authors have sufficient statistics in all size bins to state that these particles are only from 280-380 nm?**

Every size bin includes a sufficient number of particles ( >20) to conduct a statistical analysis. In terms of absolute number, "Non-K,NH$_4$-containing" TMA particles are most abundant between 280 and 420 nm which makes them distinguishable from other TMA particle sub-types. Considering relative fractions, the abundance of "Non-K,NH$_4$-containing" TMA particles is highest between 280 and 320 nm compared to the other types. We re-formulated this sentence as follows (p.16, l.8 – p.17, l.2):

This is consistent with results from particle size distributions of TMA sub-types in Fig. 10 illustrating that the fractional abundance of "Non-K,NH$_4$-containing" TMA particles is

highest between 280 and 380 nm compared to other sub-types containing levoglucosan and/or potassium.

**(49)    Page 17, Lines 4-5: If 8 July is a focus here, why did the authors choose not to show the wind data, even in the supplemental?**

We added a figure illustrating the wind speed along the flight track on July 8 to the Supplement (Fig. S9).

**(50)    Figure 14: Please define "potential emission sensitivity" in the caption or text. The units of seconds are not clear.**

We added an explanation on PES maps in Sect. 2.3 (p.5, l.20-22):

FLEXPART was operated in backward mode to provide potential emission sensitivity (PES) maps, which are the response functions to tracer releases from a receptor location. The value of the PES function is related to the particles' residence time in the output grid cell (for more details see Sect. 5 in Stohl et al. (2005) and Stohl (2006)).

Furthermore, the caption of Fig. 17 had been changed as follows:

The color-coded area presents values of the potential emission sensitivity (PES) function in a particular grid cell (Sect. 2.3).

In addition to changes suggested by the referees, we did some minor changes in the revised manuscript as follows:

**(1)** The vertically resolved fraction of different particle types in figures 6 and 11 are now cumulative presented. This improves readability and makes the comparison between different particle types easier.

**(2)** Particles summarized as "Others" appear now in figures 6 and 11.

**(3)** We unified axis notation in figures 5, 6, 10, 11, 12, 14 and 15.

**(4)** The colored flight tracks in Fig. 2 were partly wrongly assigned. We changed this.

**(5)** As described in the response to reviewer #1 comment #1, the inlet we used for aerosol sampling is not suitable for in-cloud measurements. Therefore, aerosol measurements inside clouds had been discarded. In the revised version of the manuscript, this selection had been made up for the vertical profiles (median and interquartile ranges) of $N_{d>5nm}$ and $N_{d>250nm}$ (Fig. 3).

**References:**

[revised manuscript text omitted]

---

## Referee Report (RR1)

I am providing a follow-up review to Kollner et al. "Particulate trimethylamine in the summertime Canadian high Arctic lower troposphere." I reviewed the responses to all three reviewers and believe the manuscript has been significantly improved. Notably, the attribution of TMA to biomass burning, the consideration of SPMS organic nitrogen markers, and the classification of a greater fraction of analyzed particles were particularly important revisions that were implemented. While the manuscript could be more concise, it is comprehensive, which is also important. My remaining suggestions are provided below.

Major comments:

Section 3.2: It is a significant improvement that the "K/S-containing" particle type, corresponding to an additional 29% of the particle number analyzed by ALABAMA, is now included in the manuscript. While the authors say in the responses that they intend to focus on the TMA-containing particles, this overview of all particles characterized provides context for this analysis and is currently presented in paper. Therefore, it is important to properly present an overview of these particle types, even if in-depth analysis of these particles is not conducted or presented. Please add a note about why 28% of the particles still remain unclassified, as this is still a significant fraction. The addition of the "others" mean mass spectrum as Fig S7 is quite useful; however, it is very similar to the "K/S-containing" particle type, leaving me wondering why the majority of these particles were not included with those particles. If there are differences in the intensities of the peaks, could these be due to shot-to-shot variability of the desorption/ionization laser interacting with the particles? These particles seem to have the same components in the same general mass spectral pattern, with the exception of perhaps less intense negative ion mass spectra (perhaps due to water content variations between particles). With this consideration, could a greater fraction of these "others" particles be re-classified?

Section 3.2: It would also be useful here, or in sub-section(s), to briefly comment on the particle types observed during this study compared to the previous Arctic summertime single-particle mass spectrometry studies by Sierau et al (2014, ACP) and Gunsch et al (2017, ACP). This would provide context for the variability between Arctic studies.

Discussion associated with Figs 6, 10, 11, 12, 14, & 15: Since many of the vertical, size and time bins contain fewer than 100 analyzed particles each, it would be useful to calculate and report errors, according to binomial statistics, on the discussed number fractions.

Minor suggestions/comments:

General: Since few readers have experience with both the Aerodyne AMS and single-particle mass spectrometers and this paper now shows data from both, it is very important throughout the manuscript to indicate whether fractions correspond to number or mass fractions to reduce potential confusion (also since most of the atmospheric chemistry community is used to thinking about mass concentrations).

Table 1: The clarifications to this table are very helpful. Thank you!

P12 L15: Note that magnesium and calcium are regular components of seawater; biologically-active water is not required, as implied here.

Fig. 8: For the reader not familiar with the differences between the HR-ToF-AMS and ALABAMA, I suggest pointing out in the caption that the number fractions and total numbers correspond to ALABAMA data as this could get misunderstood/lost by the non-expert reader. Perhaps also change the figure labeling with this clarification in mind.

P22 L17-20: The authors seem to contradict themselves by saying that "this is the first study demonstrating the incorporation of amines in Arctic aerosol from inner-Arctic sources" and following this by citing the work of Leaitch et al (2017) who present "measurements confirm[ing] the presence of particulate amines and its marine-biogenic source at another Arctic site".

---

## Author Response (AR2)

Dear Barbara Ervens,

we want to thank Referee #3 for her/his further comments and suggestions. Our response is formatted as follows:

**Reviewer's comments**

Author's reply

**Changes to the manuscript**

All page, line, section and figure numbers in bold refer to the original manuscript, all others to the revised version.

Section 3.2: It is a significant improvement that the "K/S-containing" particle type, corresponding to an additional 29% of the particle number analyzed by ALABAMA, is now included in the manuscript. While the authors say in the responses that they intend to focus on the TMA-containing particles, this overview of all particles characterized provides context for this analysis and is currently presented in paper. Therefore, it is important to properly present an overview of these particle types, even if indepth analysis of these particles is not conducted or presented. Please add a note about why 28% of the particles still remain unclassified, as this is still a significant fraction. The addition of the "others" mean mass spectrum as Fig S7 is quite useful; however, it is very similar to the "K/S-containing" particle type, leaving me wondering why the majority of these particles were not included with those particles. If there are differences in the intensities of the peaks, could these be due to shot-to-shot variability of the desorption/ionization laser interacting with the particles? These particles seem to have the same components in the same general mass spectral pattern, with the exception of perhaps less intense negative ion mass spectra (perhaps due to water content variations between particles). With this consideration, could a greater fraction of these "others" particles be re-classified?

Many of the remaining (unclassified) particles were single-polarity spectra (likely due to particulate water suppresses negative ion formation). Therefore, these particles had not been added previously to the K/S-containing type. However, we decided to re-classify "others" as follows: all particles containing potassium (but not necessarily sulfate) were summarized as K-containing particles. With this consideration, the new K-containing particle type includes the former dual-polarity spectra (including potassium and sulfate) (29 %), the (new) dual-polarity spectra (including potassium and no sulfate, but other anions signals, such as organic-nitrogen fragments) (8 %) and the (new) single-polarity spectra (including potassium and no indications about anion signals) (9 %). Therefore, the new group of K-containing particles replaced the former K/S-containing group and accounts for 46 % by number. Further, the remaining (unclassified) number of particles in "others" decreased from 28 % to 10 %.

According to this re-classification, figures 4e, 5, 6, 15, S7 and Tab. 2 had been changed. The K-containing particle type is presented and introduced (like the former K/S-containing group) in a mean bipolar mass spectrum (Fig. 4e), in Tab. 2 and in Sect. 3.2.1. In several text parts "K/S-containing" had been changed to "K-containing ". Further changes in the text had been done as follows:

p. 8, 1. 2-5

Applying the marker method (Sect. 2.4), we classified 6676 particle mass spectra (90 % of the mass spectra analyzed by the ALABAMA (Sect. 2.4)) into five distinct particle types: TMA-, Na/Cl-, EC-, levoglucosan- and K-containing particles. TMA-, levoglucosan- and K-containing particles, with

relative fractions of 23 %, 18 % and 46 %, respectively, appear to be the most prominent particle types.

p. 8, 1. 9-12

28 % and 9 % of TMA- and K-containing particle spectra lack negative ions, respectively. Potential reasons for the lack of negative ions are discussed in the Supplement Sect. 1. The mean spectrum of the remaining 736 particles (10 % of mass spectra analyzed by the ALABAMA), which could not be classified into one of the five particle groups outlined above, is shown in Fig. S7.

Section 3.2: It would also be useful here, or in sub-section(s), to briefly comment on the particle types observed during this study compared to the previous Arctic summertime single-particle mass spectrometry studies by Sierau et al (2014, ACP) and Gunsch et al (2017, ACP). This would provide context for the variability between Arctic studies.

This comparison had been done as follows:

**p.2, 1.32-33**

Gunsch et al. (2017, Supplement) briefly mentioned the detection of particulate TMA at a coastal Alaskan site in summer.

**p.12, 1.3-12**

Previous Arctic SPMS studies by Sierau et al. (2014) and Gunsch et al. (2017) reported a similar particle type to EC-containing particles (noted as ECOC type 1 and soot, respectively). Sierau et al. (2014) attributed this particle type to remote biomasss/biofuel sources of continental origin. In contrast, Gunsch et al. (2017) assigned a large fraction of soot particles to emissions from the nearby oil fields at Prudhoe Bay. In the recent study, the remote location of Resolute Bay excludes a larger influence of oil and gas extraction activities (Aliabadi et al., 2015; Peters et al., 2011). Further, Sierau et al. (2014) analyzed a particle type similar to the K-containing type in this study and noted as K-CN-sulfate type. They have speculated about a marine origin of these mixtures of potassium, sulfate and organic-nitrogen fragments. Sodium and MSA were partially present on the K-containing particle type in our study (Fig. 4e and Tab. 2), which confirms the hypothesis of Sierau et al. (2014). However, it is likely that this large group of K-containing particles (46 %) includes different emission sources within and above the local BL.

**p.12, 1.18-21**

Sierau et al. (2014) and Gunsch et al. (2017) did not report the detection of levoglucosan with SPMS measurements in the summertime Arctic. It is likely that these ground-based measurements missed a large fraction of particles typically present above the BL (including levoglucosan particles).

**p.13, 1.8-9**

Similar ion peaks were observed by Sierau et al. (2014) and Gunsch et al. (2017) and assigned to aged sea spray particles.

Discussion associated with Figs 6, 10, 11, 12, 14, & 15: Since many of the vertical, size and time bins contain fewer than 100 analyzed particles each, it would be useful to calculate and report errors, according to binomial statistics, on the discussed number fractions.

Errors to the associated number fraction of identified particle types (using binomial statistics) had been added to the suggested figures and Fig. 5.

General: Since few readers have experience with both the Aerodyne AMS and single-particle mass spectrometers and this paper now shows data from both, it is very important throughout the manuscript to indicate whether fractions correspond to number or mass fractions to reduce potential confusion (also since most of the atmospheric chemistry community is used to thinking about mass concentrations).

We stated once at the beginning of the discussion (p. 8, 1.19-20) that the following use of the word fraction refers to number fraction measured by ALABAMA.

**Table 1: The clarifications to this table are very helpful. Thank you!**

Thanks.

**P12 L15: Note that magnesium and calcium are regular components of seawater; biologically-active water is not required, as implied here.**

We re-formulated this sentence as follows:

It is known from previous SPMS laboratory studies on sea spray particles produced from biologically active waters that organic nitrogen species present on inorganic salts arise from biological activity (Prather et al., 2013; Guasco et al., 2014).

Fig. 8: For the reader not familiar with the differences between the HR-ToF-AMS and ALABAMA, I suggest pointing out in the caption that the number fractions and total numbers correspond to ALABAMA data as this could get misunderstood/lost by the non-expert reader. Perhaps also change the figure labeling with this clarification in mind.

We added *ALABAMA* to the label of the color-scale in Fig. 8. We further added *ALABAMA* at the appropriate place in the caption of this figure.

**P22 L17-20: The authors seem to contradict themselves by saying that "this is the first study demonstrating the incorporation of amines in Arctic aerosol from inner-Arctic sources" and following this by citing the work of Leaitch et al (2017) who present "measurements confirm[ing] the presence of particulate amines and its marine-biogenic source at another Arctic site".**

The manuscript by Leaitch et al. (2017) was published in ACPD on August 4, 2017, i.e. eight weeks later than our ACPD manuscript (June 8). Therefore, these sentences do not contradict each other.

Best regards, Franziska Köllner On behalf of all the co-authors

**Particulate trimethylamine in the summertime Canadian high Arctic lower troposphere**

Franziska Köllner1,2, Johannes Schneider1, Megan D. Willis3, Thomas Klimach1, Frank Helleis1, Heiko Bozem2, Daniel Kunkel2, Peter Hoor2, Julia Burkart3, W. Richard Leaitch4, Amir A. Aliabadi4,a, Jonathan P.D. Abbatt3, Andreas B. Herber5, and Stephan Borrmann1,2

1Max Planck Institute for Chemistry, Mainz, Germany

[revised manuscript text omitted]